# Chromatinization of *Escherichia coli* with archaeal histones

**Maria Rojec[1,2][†], Antoine Hocher[1,2][†], Kathryn M Stevens[1,2], Matthias Merkenschlager[1,2], Tobias Warnecke[1,2]\***

[1]Medical Research Council London Institute of Medical Sciences, London, United Kingdom; [2]Institute of Clinical Sciences, Faculty of Medicine, Imperial College London, London, United Kingdom

**Abstract** Nucleosomes restrict DNA accessibility throughout eukaryotic genomes, with repercussions for replication, transcription, and other DNA-templated processes. How this globally restrictive organization emerged during evolution remains poorly understood. Here, to better understand the challenges associated with establishing globally restrictive chromatin, we express histones in a naive system that has not evolved to deal with nucleosomal structures: *Escherichia coli*. We find that histone proteins from the archaeon *Methanothermus fervidus* assemble on the *E. coli* chromosome in vivo and protect DNA from micrococcal nuclease digestion, allowing us to map binding footprints genome-wide. We show that higher nucleosome occupancy at promoters is associated with lower transcript levels, consistent with local repressive effects. Surprisingly, however, this sudden enforced chromatinization has only mild repercussions for growth unless cells experience topological stress. Our results suggest that histones can become established as ubiquitous chromatin proteins without interfering critically with key DNA-templated processes.
DOI: https://doi.org/10.7554/eLife.49038.001

**\*For correspondence:**
tobias.warnecke@imperial.ac.uk

[†]These authors contributed equally to this work

**Competing interests:** The authors declare that no competing interests exist.

## Introduction

All cellular systems face the dual challenge of protecting and compacting their resident genomes while making the underlying genetic information dynamically accessible. In eukaryotes, this challenge is solved, at a fundamental level, by nucleosomes,~147 bp of DNA wrapped around an octameric histone complex. Nucleosomes can act as platforms for the recruitment of transcriptional silencing factors such as heterochromatin protein 1 (HP1) in animals (*Danzer and Wallrath, 2004*; *Zhao et al., 2000*) and Sir proteins in yeast (*Gartenberg and Smith, 2016*), but can also directly render binding sites inaccessible to transcription factors (*Beato and Eisfeld, 1997*; *Zhu et al., 2018*). As a consequence, gene expression in eukaryotes is often dependent on the recruitment of chromatin remodelers. By controlling access to DNA, histones play a key role in lowering the basal rate of transcription in eukaryotic cells and have therefore been described as the principal building blocks of a restrictive transcriptional ground state (*Struhl, 1999*).

Histones are not confined to eukaryotes, but are also common in archaea (*Adam et al., 2017*; *Henneman et al., 2018*). They share the same core histone fold but typically lack N-terminal tails, which are the prime targets for post-translational modifications in eukaryotes (*Henneman et al., 2018*). As tetrameric complexes, they wrap ~60 bp instead of ~147 bp of DNA (*Reeve et al., 2004*). At least in some archaea, these tetrameric complexes can be extended, in dimer steps, to form longer oligomers that wrap correspondingly more DNA (~90 bp,~120 bp, etc.) and assemble without the need for dedicated histone chaperones (*Xie and Reeve, 2004*; *Mattiroli et al., 2017*; *Maruyama et al., 2013*). Archaeal and eukaryotic nucleosomes preferentially assemble on DNA that is more bendable, a property associated with elevated GC content and the presence of certain periodically spaced dinucleotides, notably including AA/TT (*Ammar et al., 2011*; *Nalabothula et al.,*

*2013*; *Pereira et al., 1997*; *Bailey et al., 2000*; *Ioshikhes et al., 2011*). They also exhibit similar positioning around transcriptional start sites (*Ammar et al., 2011*; *Nalabothula et al., 2013*), which are typically depleted of nucleosomes and therefore remain accessible to the core transcription machinery. Whether archaeal histones play a global restrictive role akin to their eukaryotic counterparts, however, remains poorly understood, as does their involvement in transcription regulation more generally (*Gehring et al., 2016*).

Thinking about the evolution of restrictive chromatin and its molecular underpinnings, we wondered how the presence of histones would affect a system that is normally devoid of nucleosomal structures. How would a cell that has neither dedicated nucleosome remodelers nor co-evolved sequence context cope with chromatinization? Could global chromatinization occur without fundamentally interfering with DNA-templated processes? How easy or hard is it to transition from a system without histones to one where histones are abundant? What are the key adaptations required, if any, to accommodate histones?

Motivated by these questions, we built *Escherichia coli* strains expressing histones from the hyperthermophilic archaeon *Methanothermus fervidus* (HMfA or HMfB), on which, thanks to the pioneering work of Reeve and co-workers, much of our foundational knowledge about archaeal histones is based. HMfA and HMfB are 85% identical at the amino acid level but differ with regard to their DNA binding affinity and expression across the *M. fervidus* growth cycle, with HMfB more prominent toward the latter stages of growth and able to provide greater DNA compaction in vitro (*Sandman et al., 1994*; *Marc et al., 2002*). We find that HMfA and HMfB, heterologously expressed in *E. coli*, bind to the *E. coli* genome and protect it from micrococcal nuclease (MNase) digestion, allowing us to map nucleosomes in *E. coli* in vivo. We present evidence for sequence-dependent nucleosome positioning and occupancy and consider how the presence of histones affects transcription on a genome-wide scale. Importantly, we find evidence for local repressive effects associated with histone occupancy yet only mild repercussions for growth and cell morphology, unless cells are forced to deal with excess levels of DNA damage or topological stress. Under favourable conditions, *E. coli* copes remarkably well with enforced chromatinization, despite evidence that histones disrupt the binding of native nucleoid-associated proteins (NAPs). Our findings have implications for how histones became established as global repressive regulators during the evolution of eukaryotes and for the evolvability of transcriptional ground states.

## Results

### Archaeal histones bind the *E. coli* genome in vivo, assemble into oligomers, and confer protection from MNase digestion

We transformed an *E. coli* K-12 MG1655 strain with plasmids carrying either *hmfA* or *hmfB*, codon-optimised for expression in *E. coli* and under the control of a rhamnose-inducible promoter (see Materials and methods, *Figure 1—figure supplement 1*). Below, we will refer to these strains as Ec-hmfA and Ec-hmfB, respectively, with Ec-EV being the empty vector control strain (*Supplementary file 1*). Following induction, both histones are expressed at detectable levels and predominantly found in the soluble fraction of the lysate in both exponential and stationary phase (*Figure 1—figure supplement 2*). We did not observe increased formation of inclusion bodies. Based on dilution series with purified histones (see Materials and methods, *Figure 1—figure supplement 2*), we estimate HMfA:DNA mass ratios of up to ~0.6:1 in exponential (~0.7:1 in stationary phase), which corresponds to one histone tetramer for every 76 bp (64 bp) in the *E. coli* genome. Given that a tetramer wraps ~60 bp of DNA, this implies a supply of histones that is, in principle, sufficient to cover most of the *E. coli* genome. However, it is important to note that, at any given time, not all histones need to be associated with DNA.

We carried out MNase digestion experiments using samples from late exponential and stationary phase, corresponding to 2 hr and 16–17 hr after induction, respectively (see Materials and methods). In response to a wide range of enzyme concentrations, MNase digestion of chromatin from Ec-hmfA/B (see Materials and methods) yields a ladder-like pattern of protection that is not observed in Ec-EV (*Figure 1A–B*). Across many replicates, we could usually discriminate the first four rungs of the ladder, with the largest rung at 150 bp. On occasion, we observe multiple larger bands (e.g. for Ec-hmfA in *Figure 1A*). Sequencing digestion fragments < 160 bp using single-end Illumina

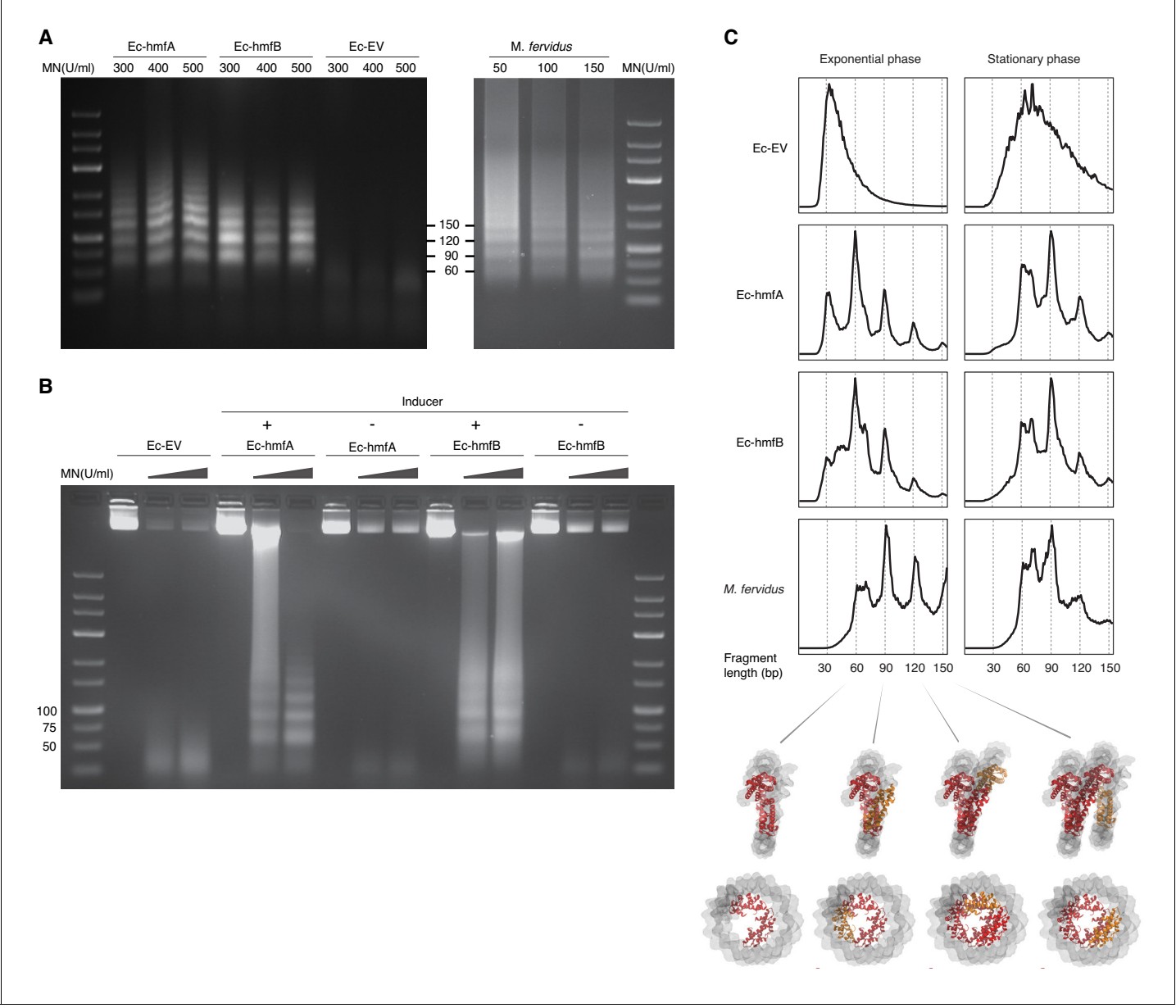

**Figure 1.** MNase digestion of *M. fervidus* and *E. coli* strains expressing *M. fervidus* histones. (A) Agarose gel showing profiles of DNA fragments that remain protected at different MNase (MN) concentrations. (B) Ladder-like protection profiles are only observed when hmfA/B expression is induced. (C) Length distribution profiles of sequenced fragments show peaks of protection at multiples of 30 bp in histone-expressing strains. Structural views below highlight how these 30 bp steps would correspond to the addition or removal of histone dimers, starting from the crystal structure of a hexameric HMfB complex (PDB: 5t5k), which wraps ~90 bp of DNA.

DOI: https://doi.org/10.7554/eLife.49038.002

The following figure supplements are available for figure 1:

**Figure supplement 1.** Layout of pD681-derived plasmids used in this study.

DOI: https://doi.org/10.7554/eLife.49038.003

**Figure supplement 2.** Detection and quantification of HMf expression in *E. coli*.

DOI: https://doi.org/10.7554/eLife.49038.004

technology recapitulates the read length distribution seen on gels, with peaks around 60 bp, 90 bp, 120 bp, and 150 bp (*Figure 1C*), consistent with oligomerization dynamics described for archaeal histones in their native context (*Maruyama et al., 2013*; *Mattiroli et al., 2017*). Indeed, we obtained remarkably similar digestion profiles when we applied the same protocol, modified to account for altered lysis requirements (see Materials and methods), to *M. fervidus* cultures (*Figure 1A,C*). Modal fragment sizes of ~60 bp and ~90 bp in exponential and stationary phase (*Figure 1C*), respectively, suggest that larger oligomers become more prevalent later in the growth cycle, which might reflect elevated histone:DNA ratios but also reduced perturbation from replication and transcription, as further discussed below. In exponential phase only, an additional peak is evident at ~30 bp. Fragments of this size were previously observed during in vitro reconstitution experiments with HMfA/B and, at the time, attributed to the binding of histone dimers (*Grayling et al., 1997*). However, in our digestion regime, this peak is also present in Ec-EV, and we cannot therefore rule out the possibility that it is caused by specifics of the digestion protocol, library construction or native *E. coli* proteins found exclusively in exponential phase. Below, we therefore focus on larger peaks (60 bp, 90 bp, etc.) that are absent from Ec-EV, but present in *M. fervidus* and our histone-bearing *E. coli* strains.

## Intrinsic sequence preferences govern nucleosome formation along the *E. coli* genome

Mapping digestion fragments to the *E. coli* genome, we find that binding is ubiquitous. On a coarse scale, coverage across the chromosome appears relatively even (*Figure 2A*). On a more local scale, however, protected fragments group into defined binding footprints (*Figure 2C*). Local occupancy (measured for 60 bp windows, overlapping by 30 bp) is highly correlated across replicates (*Figure 2D*), consistent with non-random binding. Ec-hmfA and Ec-hmfB are also highly correlated (*Figure 2E*); minor differences may reflect subtly different binding preferences, as previously reported (*Bailey et al., 2000*). Areas of apparent histone depletion often coincide with AT-rich domains (*Figure 2C,F*): nucleosomes are depleted from AT-rich transcriptional start sites (TSSs), mimicking a key aspect of nucleosome architecture in eukaryotes and archaea (*Figure 2G*), and extension into longer oligomers is less likely when tetramer binding footprints are flanked by AT-rich sequence (*Figure 2—figure supplement 1*), as is the case in *M. fervidus* (*Hocher et al., 2019*).

The above observations point to a role for sequence composition in determining nucleosome positioning and/or occupancy but likely also reflect known MNase preferences for AT-rich DNA (see Ec-EV in *Figure 2G* in particular). To discriminate between these two factors, we first analysed read-internal nucleotide enrichment patterns, which should be unaffected by MNase bias. Considering fragments of exact size 60 bp (90 bp, etc. see Materials and methods), we find dyad-symmetric nucleotide enrichment patterns that are absent from size-matched Ec-EV fragments but mirror what is seen in fragments from native *M. fervidus* digests (*Figure 3A*), despite large differences in overall genomic GC content. Next, to disentangle conflated signals of MNase bias and nucleosomal sequence preferences directly, and to assess their relative impact on inferred occupancy across the genome, we normalized coverage in Ec-hmfA/B by coverage in Ec-EV (see Materials and methods). We then trained LASSO models for different fragment size classes (60 bp, 90 bp, 120 bp) to predict normalized occupancy across the genome from the underlying sequence, considering all mono-, di-, tri-, and tetra-nucleotides as potential predictive features (see Materials and methods, *Supplementary file 2*). We find that sequence is a good predictor of normalized occupancy in stationary phase (*Figure 3B–C*), particularly for larger fragments (e.g. 120 bp footprints in Ec-hmfA: $\rho = 0.72$, $p<2.2\times10^{-16}$; 120 bp footprints in Ec-hmfB: $\rho = 0.76$, $p=<2.2\times10^{-16}$, *Figure 3C*). GC content as a simple metric captures much of the variability in occupancy (*Figure 3B,D*).

Interestingly, however, the predictive power of sequence is dramatically reduced in exponential phase (*Figure 3B,D*). Why would this be? We suspect that stationary phase represents a comparatively more settled state, characterized by reduced replication, transcription, and other DNA-templated activity, that is more conducive to the establishment or survival of larger oligomers and where nucleosome formation is better able to track intrinsic sequence preferences. In support of this hypothesis, we find that transcriptional activity modulates the relationship between GC content and occupancy: the relationship is stronger where transcriptional activity is weaker ($\rho = -0.46$, $p=0.039$; *Figure 3E*). Importantly, this does not imply that higher transcription leads to reduced histone occupancy. In fact, there is no negative correlation between transcript levels in Ec-EV and histone occupancy (*Figure 3F*, $\rho >0.1$ for all growth phase/histone combinations). Rather, these results are

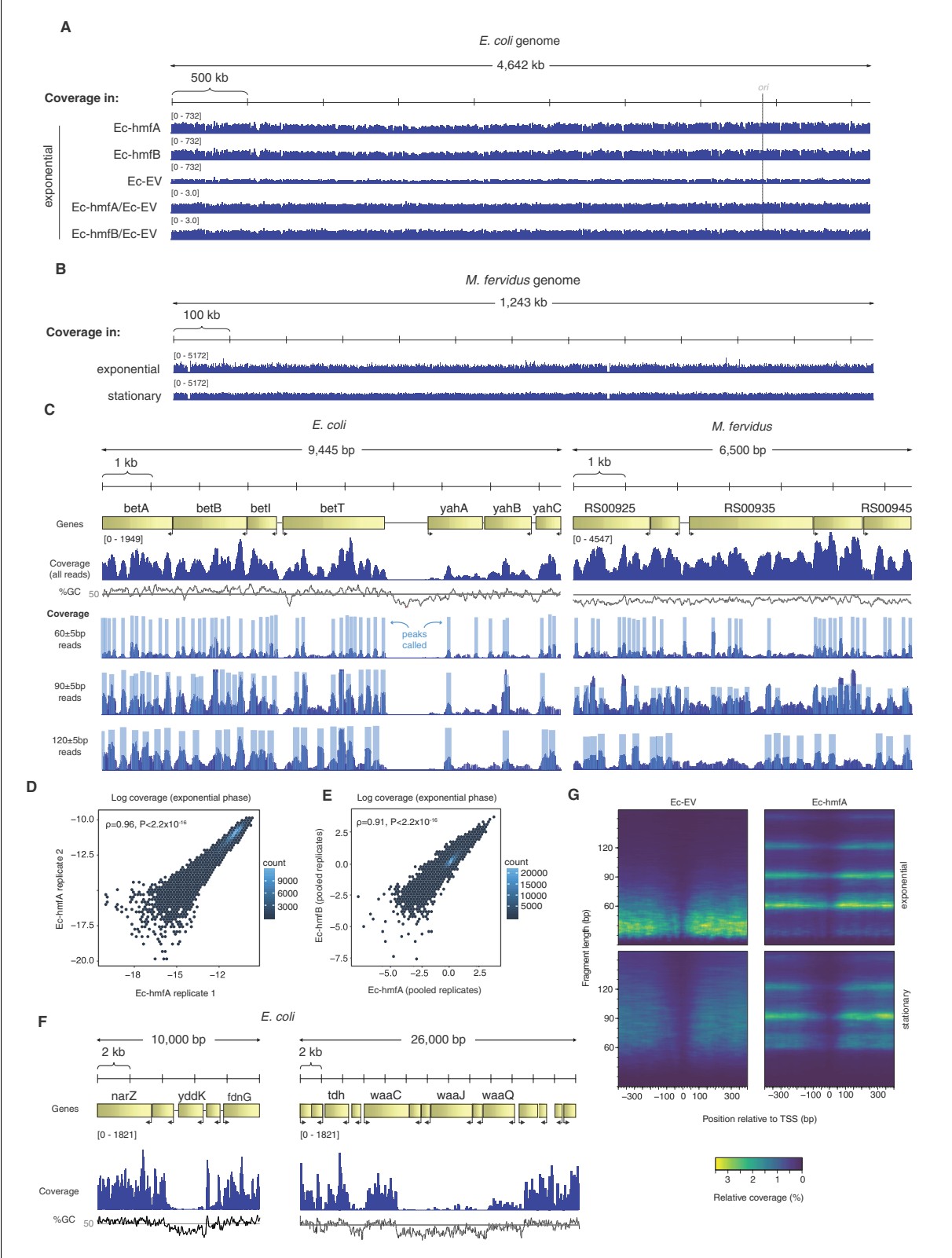

**Figure 2.** Distribution of MNase-protected fragments across the *E. coli* genome. (**A**) Genome-wide coverage (and normalized coverage) tracks of MNase-protected fragments along the *E. coli* K-12 MG1655 and (**B**) the M. *fervidus* genome. (**C**) Fragments of defined size cluster into footprints in *E. coli* and *M. fervidus,* as illustrated for two example regions. (**D**) Correlation in coverage measured for two biological replicates of Ec-hmfA. Coverage here is expressed as a proportion of total reads in a given replicate. (**E**) Correlation in normalized coverage between Ec-hmfA and Ec-hmfB. Reads were

*Figure 2 continued on next page*

*Figure 2 continued*

pooled across replicates for each strain. (F) Two examples from Ec-hmfA highlighting that drops in coverag frequently correspond to regions of low GC content. (G) Coverage as a function of both distance from experimentally defined transcriptional start sites (see Materials and methods) and fragment size.

DOI: https://doi.org/10.7554/eLife.49038.005

The following figure supplement is available for figure 2:

**Figure supplement 1.** Sequence-dependent oligomer extension dynamics.

DOI: https://doi.org/10.7554/eLife.49038.006

consistent with transcription increasing the fuzziness of nucleosome positioning. We also find a better correlation between sequence composition and occupancy further away from the origin of replication, suggestive of replication-associated perturbation (*Figure 3G*).

## Evidence that nucleosome formation locally represses transcription

Next, we asked whether the presence of histones in *E. coli* affects transcription. We first consider whether histones exert direct repressive effects in cis. Further below, we look at genome-wide transcriptional responses to histone expression more broadly to understand how *E. coli* is challenged by and adapts to the presence of histone proteins.

To address the first question, we generated two additional strains, Ec-hmfA$_{nb}$ and Ec-hmfB$_{nb}$, where *hmfA* and *hmfB*, respectively, were recoded to carry three amino acid changes (K13T-R19S-T54K) previously shown to abolish DNA binding of HMfB (*Soares et al., 2000*). MNase treatment of these strains resulted in digestion profiles similar to Ec-EV, consistent with compromised ability to form protective nucleosomal structures (*Figure 4—figure supplement 1*). Using RNA-Seq, we quantified differential transcript abundance in Ec-hmfA versus Ec-EV and Ec-hmfA$_{nb}$ versus Ec-EV (see Materials and methods) and then excluded genes from further analysis that were significantly up-regulated (or down-regulated) in both comparisons, reasoning that coincident patterns of change are not uniquely attributable to binding and might instead derive from systemic responses to heterologous expression. We then considered differential expression in Ec-hmfA/B versus Ec-EV for the remaining genes as a function of nucleosome occupancy.

Looking at normalized coverage across gene bodies, annotated promoters and experimentally mapped transcriptional start sites, we find evidence for nucleosome-mediated dampening of transcriptional output. Notably, genes that are significantly ($P_{adj}$ <0.05) down-regulated in histone-bearing strains display significantly higher nucleosome occupancy at TSSs than upregulated genes (*Figure 4A*). This is true regardless of whether we consider occupancy at a single base assigned as the TSS, occupancy in a ± 25 bp window around that site, or occupancy across annotated promoters (see Materials and methods). This signal is lost almost entirely when considering a promoter-proximal 51 bp control window centred on the start codon (*Figure 4—figure supplement 2*). This finding argues against a model where histone occupancy increases as a consequence of downregulation. Under such a model, we would have predicted histone occupancy to increase not only at the promoter but also downstream of it. The relationship between transcriptional changes and average histone occupancy across the gene body is more complex; weaker effects in the expected direction are evident for Ec-hmfA but not Ec-hmfB (*Figure 4—figure supplement 2*).

Interestingly, repressive effects at TSSs in particular appear to be driven by larger oligomeric nucleosomes (90 bp, 120 bp, 150 bp, *Figure 4B*, *Figure 4—figure supplement 2*). This might be because larger oligomeric complexes are intrinsically more stable (*Figure 4—figure supplement 3*), harder to bypass/displace, and therefore more significant barriers to transcription initiation and elongation. In analogy to H-NS, larger oligomers might also, from an initial point of nucleation, extend to cover sequences that disfavour nucleation – a property that might facilitate promoter occlusion (*Henneman et al., 2018*; *Hocher et al., 2019*).

## Histone binding is associated with mild phenotypic effects under favourable conditions

Despite evidence for repressive effects, gross cell morphology and growth rate appear surprisingly normal. Histone-expressing cells are longer than Ec-EV cells, particularly in stationary phase, but

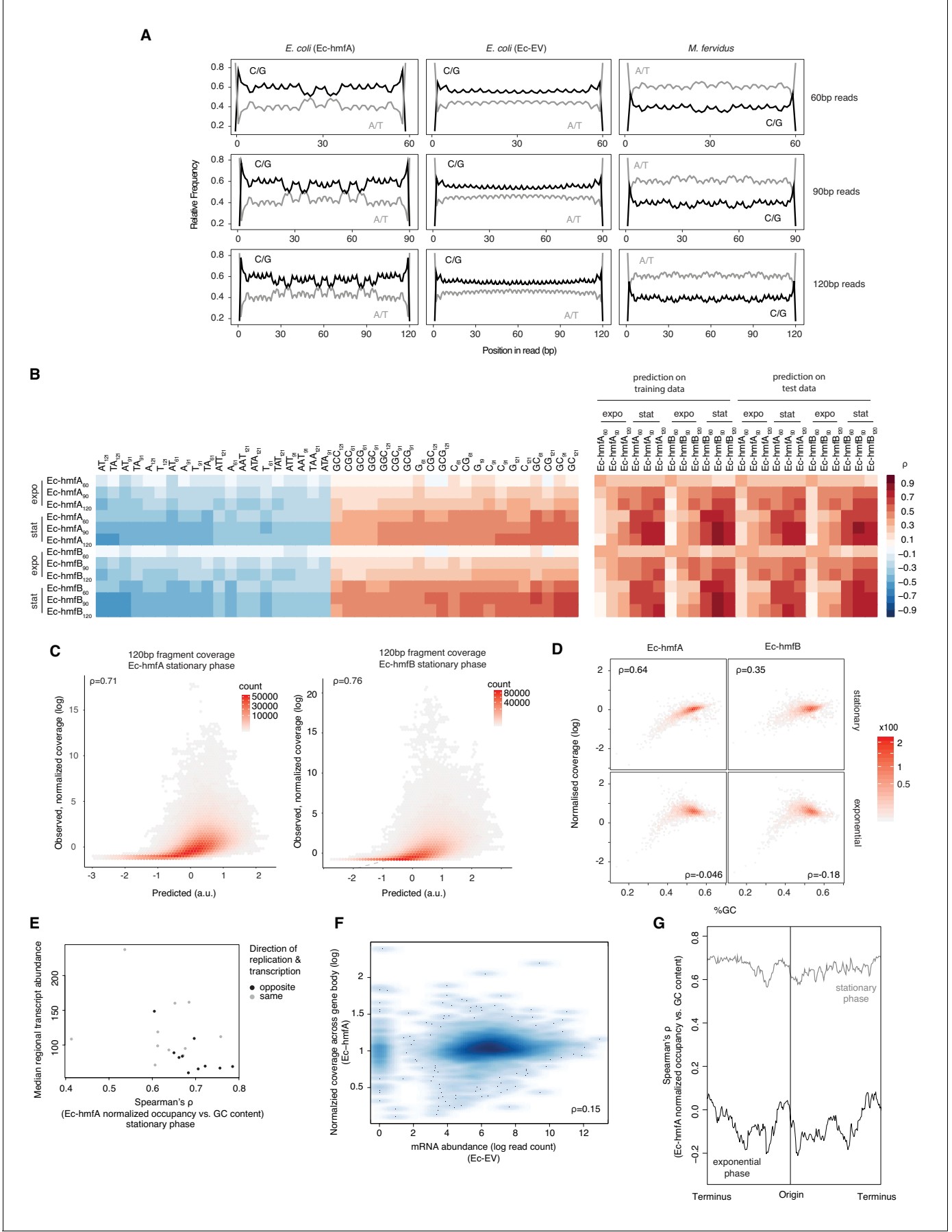

**Figure 3.** Sequence and other predictors of histone occupancy in *E. coli*. (A) Read-internal nucleotide enrichment profiles for reads of exact length 60/90/120 bp. Symmetric enrichments are evident for Ec-hmfA and *M. fervidus* native fragments but not Ec-EV. (B) Left panel: top and bottom 20 individually most informative k-mers to predict fragment size-specific normalized histone occupancy in different strains. Red and blue hues indicate positive and negative correlations between k-mer abundance and normalized occupancy, respectively. Right panel: performance of the full LASSO model on training and test data (see Materials and methods). expo: exponential phase; stat: stationary phase. (C) Correlations between predicted and observed coverage of 120 ± 5 bp fragments predicted at single-nucleotide resolution across the genome. All p<0.001. (D) GC content and normalized coverage are positively correlated in stationary but not exponential phase. All p<0.001. Coverage and GC content are measured by gene. (E) The correlation between GC content and occupancy is stronger in genomic regions where transcriptional output is lower. Regional transcriptional output is computed as median transcript abundance in a 200-gene window. To assess potential interactions between replication and transcription, windows are computed separately for genes where the directions of transcription and replication coincide and those where they differ. (F) There is no negative correlation between mRNA abundance in Ec-EV and normalized histone occupancy in Ec-hmfA, suggesting that low levels of transcription do not facilitate higher occupancy. (G) The strength of the correlation between GC content and occupancy varies along the *E. coli* chromosome. Correlations are computed for 500 neighbouring genes using a 20-gene moving window.

DOI: https://doi.org/10.7554/eLife.49038.007

they do not exhibit an altered nucleoid/cytoplasm ratio and, following a transient reduction in growth rate after induction, appear to divide normally (*Figure 5*, *Figure 5—figure supplement 1*). Under favourable conditions, growth of histone-expressing *E. coli* appears remarkably unremarkable. But how do these strains respond to stress? To find out, we monitored growth in response to

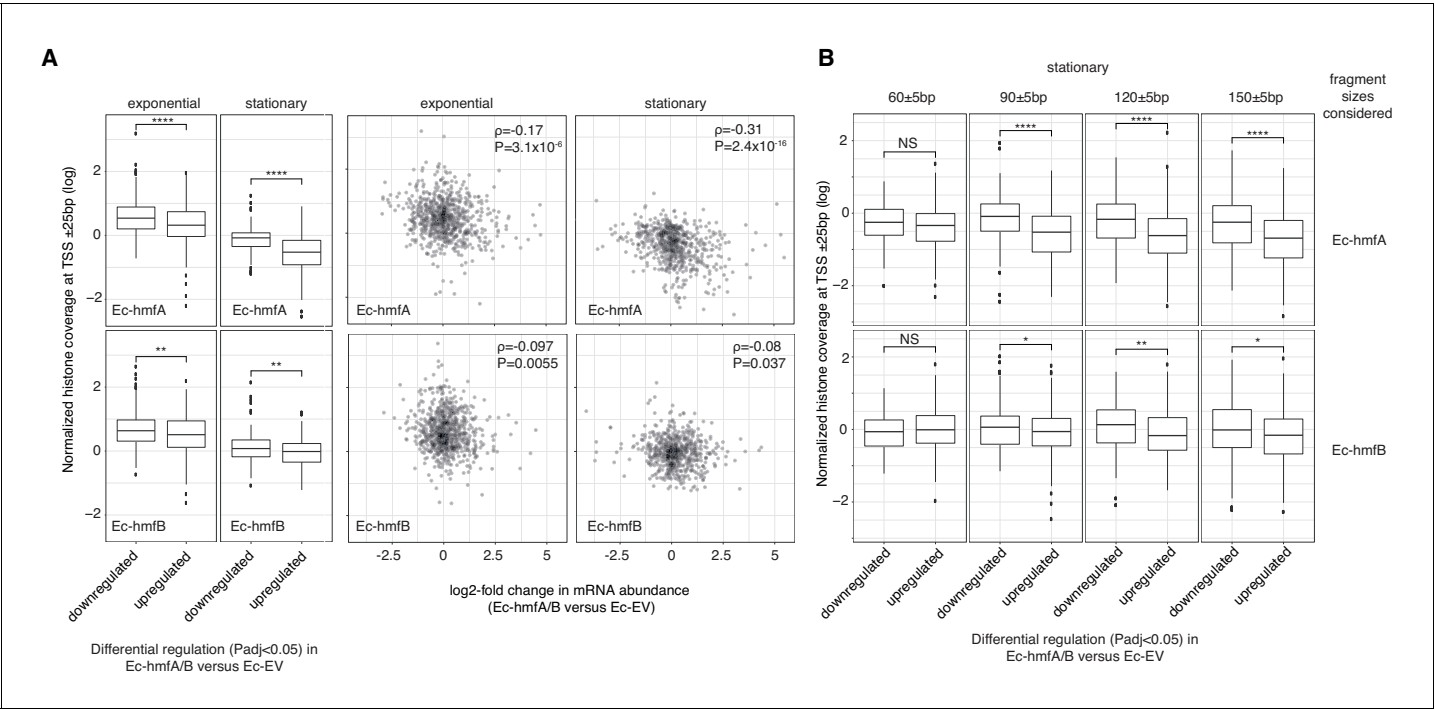

**Figure 4.** The impact of archaeal histones on transcription in *E. coli*. (A) Reduced transcript abundance in histone-expressing strains is associated with higher average histone occupancy at the TSS. Top panels: Ec-hmfA. Bottom panels: Ec-hmfB (B) Genes that are significantly downregulated in histone-expressing strains exhibit higher coverage of large (90+bp) but not small (60 bp) fragments. Top panels: Ec-hmfA. Bottom panels: Ec-hmfB. ****p<0.001; ***p<0.005; **p<0.01; *p<0.05.

DOI: https://doi.org/10.7554/eLife.49038.008

The following figure supplements are available for figure 4:

**Figure supplement 1.** Expression of non-binding histone mutants.

DOI: https://doi.org/10.7554/eLife.49038.009

**Figure supplement 2.** The impact of archaeal histones in *E. coli* on transcription.

DOI: https://doi.org/10.7554/eLife.49038.010

**Figure supplement 3.** Longer oligomeric histone-DNA complexes are more stable and have higher DNA affinity.

DOI: https://doi.org/10.7554/eLife.49038.011

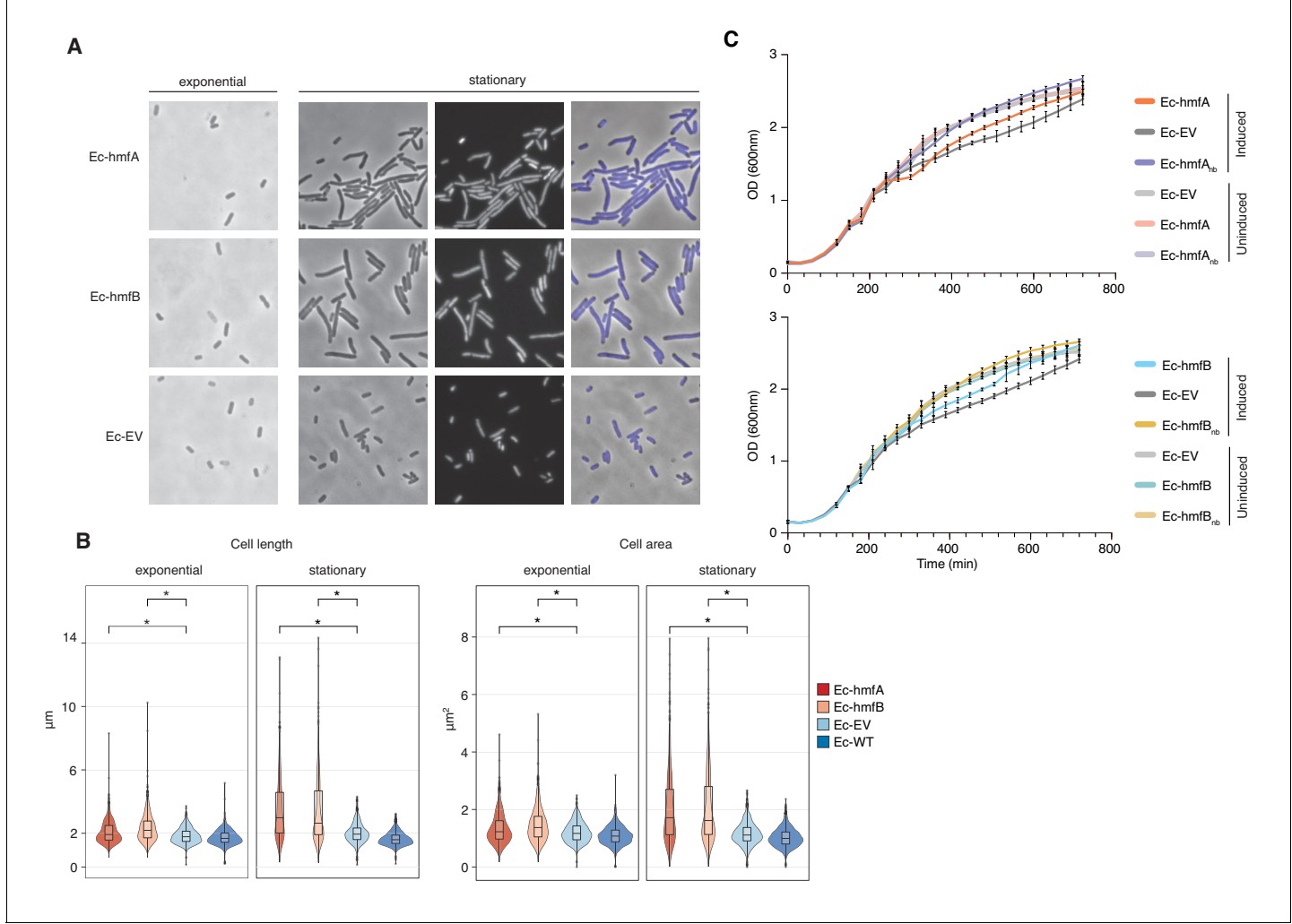

**Figure 5.** The impact of archaeal histones on *E. coli* growth. (**A**) Morphological changes triggered by HMfA and HMfB expression. Compared to the empty vector control, Ec-hmfA and Ec-hmfB become significantly longer, particularly toward the final stage of the cell cycle. DAPI staining suggests that the increase in cell length is not due to impaired cell division. Magnification 100x. (**B**) Quantification of cell length and area in histone-expressing and control strains. Some unexpectedly low values are likely attributable to debris being misidentified as cells. *p<0.0001. (**C**) Growth curves for induced and uninduced histone-expressing and control strains. Rhamnose was added for induction at 200 min.

DOI: https://doi.org/10.7554/eLife.49038.012

The following figure supplement is available for figure 5:

**Figure supplement 1.** No evidence for altered nucleoid/cytoplasm ratio in histone-expressing cells.

DOI: https://doi.org/10.7554/eLife.49038.013

transcriptional stress (rifampicin), oxidative stress (H$_2$O$_2$), DNA damage (UV), and supercoiling stress (novobiocin). To capture effects of histone occupancy during lag phase and ensure that stress responses are measured in cells where histones are established, we inoculated new cultures with cells that had already been expressing histone genes for 2 hr (see Materials and methods). When these pre-induced cells are re-inoculated, we observe a slightly prolonged lag phase (*Figure 6A*). However, histone-expressing strains recover quickly to catch up with non-binding/EV control strains. Lag phase is extended further in strains treated with rifampicin or H$_2$O$_2$ (*Figure 6A*). Again, histone-expressing strains recover well. Under these conditions, histones have a mild bacteriostatic but no bactericidal effect. In contrast, the presence of histones clearly affects the ability of cells to respond to UV and novobiocin treatment: colony formation and growth, respectively, are severely affected (*Figure 6A–B*). In novobiocin-treated histone-expressing cells, we also observe marked morphological changes, as cells become conspicuously elongated (*Figure 6C*).

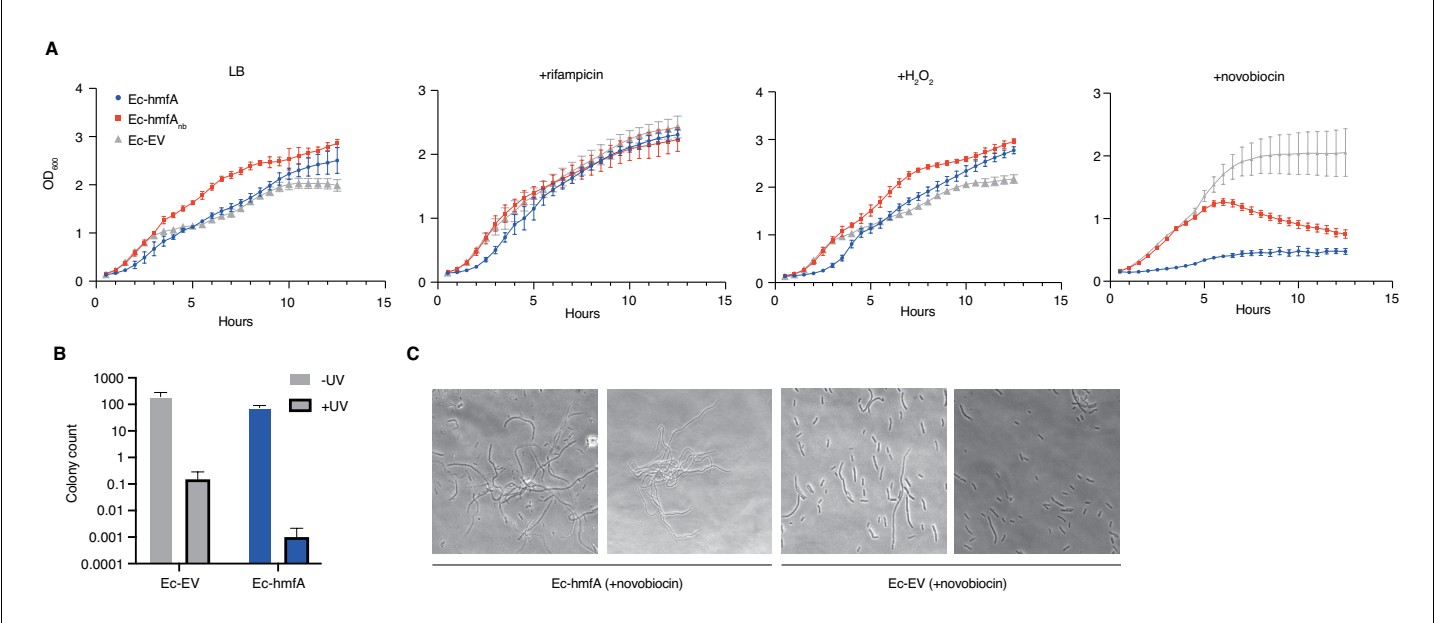

**Figure 6.** Growth responses of histone-expressing *E. coli* strains under stress. (**A**) Growth curves for pre-induced histone-expressing *E. coli* strains and controls in LB medium and LB medium with added rifampicin, $H_2O_2$, or novobiocin. See Materials and methods for growth/induction protocol and drug/chemical concentrations. (**B**) Colony counts for *E. coli* strains exposed to UV radiation or left untreated (all $p < 0.05$). (**C**) Novobiocin treatment of Ec-hmfA results in a strong filamentation phenotype.

DOI: https://doi.org/10.7554/eLife.49038.014

## Systemic transcriptional responses to histone expression in *E. coli*

The results above suggest that histones do not compromise dynamic responses to stress in general but that their presence is problematic when sensing or dealing with altered DNA topology or damage. To better understand the molecular basis of altered growth, we compared the transcriptome-wide signature of differential expression in Ec-hmfA (versus Ec-hmfA_nb, exponential phase) to >950 previously published differential expression profiles from a broad range of perturbations (see Materials and methods).

Calculating dot products as a measure of similarity between two differential expression vectors (see Materials and methods), we find that correlations between expression profiles is modest (maximum $\rho = 0.34$), indicating that the transcriptional response to histone expression has a strong unique component. Histone-expressing strains are most similar to perturbations that are marked by transient growth arrest and induction of the stringent response (amino acid starvation, cadmium shock, heat stress, *Figure 7A*, source data file 1) and to growth under metabolically challenging conditions, that is conditions where carbon sources are either scarce (stationary phase, minimal media) or suddenly altered (glucose-to-lactose shift, *Figure 7A*). Specific similarities include the downregulation of flagellar genes – a hallmark of the stringent response – and upregulation of the general stress response (RpoS regulon, *Figure 7B*). These transcriptional signatures are very much in line with the mild bacteriostatic growth phenotype (extended lag phase) we observed (*Figure 6*). Cells delay division until they have had sufficient time to adjust and even though stress responses are induced, these are not necessarily required for survival (*Figure 7—figure supplement 1*).

Downregulation of gyrases (*gyrA/B*, *Figure 7—figure supplement 1*), which introduce negative (or relax positive) supercoils, might be part of such an adaptive readjustment. Histones wrap DNA in negatively constrained supercoils so reducing gyrase expression might counteract histone-associated build-up of negative supercoiling. This might provide a quick fix, but at the cost of rendering cells more susceptible to novobiocin. In line with this idea, histone-expressing strains share transcriptional similarities to cells expressing CcdB, a gyrase poison (*Figure 7A*).

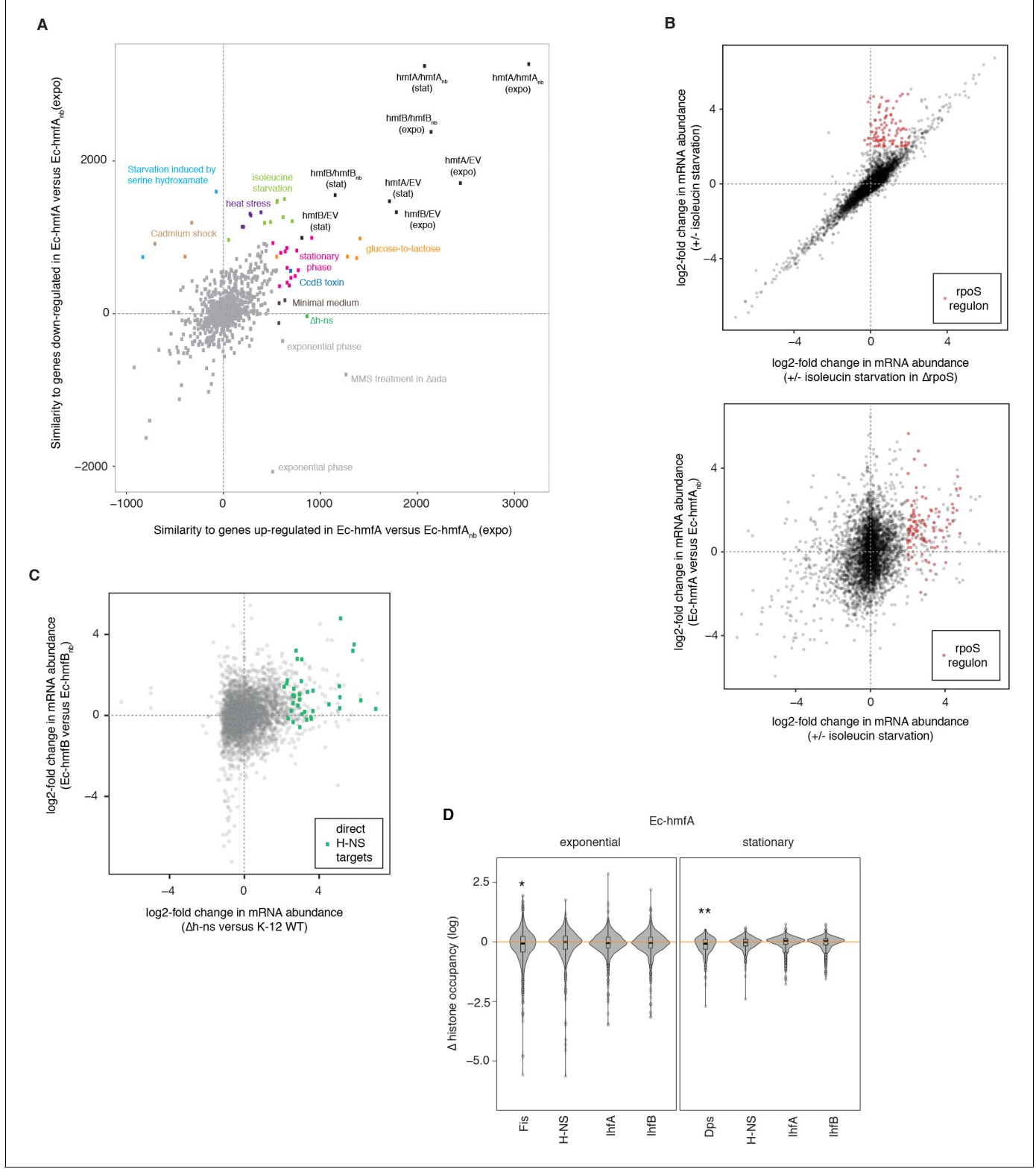

**Figure 7.** Global transcriptional responses in histone-expressing strains highlight effects on *E. coli* physiology and native chromatin organization. (**A**) Comparative analysis of global transcriptional responses, comparing up- or down-regulated genes in Ec-hmfA (versus Ec-hmfAnb) to other perturbations (underlying data provided as *Figure 7A* – source data). Perturbations with high similarity to Ec-hmfA versus Ec-hmfAnb along at least one dimension are highlighted and coloured according to the nature of the perturbation. Values < 0 indicate overall dissimilarity, equivalent to a negative correlation coefficient between the transcriptional responses. Note that the absolute similarity values here have no intrinsic meaning; only the relative

*Figure 7 continued on next page*

*Figure 7 continued*

distance from the maximum, hmfA/hmfA$_{nb}$ (expo), is meaningful. Note also that similarity should only be interpreted in reference hmfA/hmfA$_{nb}$ (expo). Points labelled 'exponential phase' constitute rare cases where, in the original study, differential expression was assessed as expo/stat rather than the more common stat/expo. When flipped, these fall into or close to the pink cluster of stationary phase datasets. (**B**) Genes controlled by RpoS (identified by comparing the response to isoleucine starvation in WT and Δ*rpoS* cells, upper panel) are upregulated upon isoleucin starvation but also in histone-expressing strains (illustrated for Ec-hmfA in the lower panel). Based on GSE11087 as provided in GenexpDB. (**C**) Correspondence between transcriptomic changes in Ec-hmfB versus Ec-hmfBnb and a Δ*h-ns* strain (GSE123554). Direct H-NS targets, as inferred by *Gawade et al. (2019)*, are highlighted in green. (**D**) Histone occupancy in regions previously found to be bound or unbound by a particular nucleoid-associated protein in *E. coli*. Δ histone occupancy is defined as the difference in histone occupancy in a region bound by a given NAP and the nearest unbound region downstream. Negative Δ(histone occupancy) values therefore indicate greater histone occupancy in areas not bound by the focal NAP, suggestive of competition for binding or divergent binding preferences. *p<0.005 **p<0.001.

DOI: https://doi.org/10.7554/eLife.49038.015

The following source data and figure supplement are available for figure 7:

**Source data 1.** Similarity to transcriptional responses observed in previous perturbations.
DOI: https://doi.org/10.7554/eLife.49038.017
**Figure supplement 1.** The impact of archaeal histones on transcription in *E. coli*.
DOI: https://doi.org/10.7554/eLife.49038.016

## Evidence that histones interfere with the binding of native nucleoid-associated proteins

We were further intrigued to see that, specifically with regard to upregulated genes, the effect of histones is similar to deleting *h-ns* (ρ = 0.19, p<2.2×10$^{-16}$). Most notably, genes previously identified as direct H-NS targets (green icons in *Figure 7C*) are amongst the most upregulated genes not only when *h-ns* is deleted (as one would expect), but also upon HMf expression. This might indicate that histones displace H-NS, but fail to provide similar silencing, leading to de-repression of H-NS target genes. In line with this hypothesis, we find that histone occupancy is not significantly reduced at known binding footprints of H-NS (*Kahramanoglou et al., 2011*), indicating that histones successfully compete for binding at those sites (*Figure 7D*). In addition to de-repression of its usual target genes, the release of H-NS might also cause gain-of-function effects, for example through the binding of AT-rich promoters that would normally not be silenced. It is interesting to note in this context that strong (>40 fold) overproduction of H-NS has previously been reported to trigger a transient (several-hour) growth arrest after which cells resume growth (*McGovern et al., 1994*). This situation, which the authors dubbed 'artificial stationary phase', is qualitatively reminiscent of the prolonged lag phase we observe upon HMf expression.

We also find little, if any, evidence for competitive exclusion at known binding sites of other endogenous NAPs (*Figure 7D*). In contrast to Δ*h-ns*, however, transcriptional responses in Δ*hupA/hupB*, Δ*dps*, and Δ*fis* strains are uncorrelated to those in Ec-hmfA/B (all ρ<|0.04|).

The above results suggest that histones readily invade genomic real estate normally occupied by endogenous NAPs. Might histones therefore, in some instances, complement NAP deletions? To address this question, we examined the effects of HMfA expression on growth in a small collection of NAP deletion strains, using the larger YFP protein as a conservative control for the burden of gratuitous protein expression. Note first that NAP deletions in *E. coli* are not associated with a strong growth phenotype, with the notable exception of the *hupA/hupB* double deletion (ΔΔHU) strain, which grows notably more slowly compared to its C600 wild-type progenitor (*Figure 8B*). HMfA expression generally leads to an increase in lag phase duration, operationally defined as the time to maximum growth rate (*Figure 8A*). This is particularly pronounced when *fis* is deleted and – for unknown and hard to interpret reasons – in M182, the wild-type progenitor strain of Δ*h-ns*. HMfA expression is also associated with a small but consistent increase in doubling time. However, in most cases, this effect is not compounded by deleting the focal NAP. The exception, again, is HU. Growth retardation associated with *hupA/hupB* deletion and HMfA expression are not additive, suggesting that histone expression might partially alleviate defects associated with the absence of HU, perhaps because both proteins constrain negative supercoils.

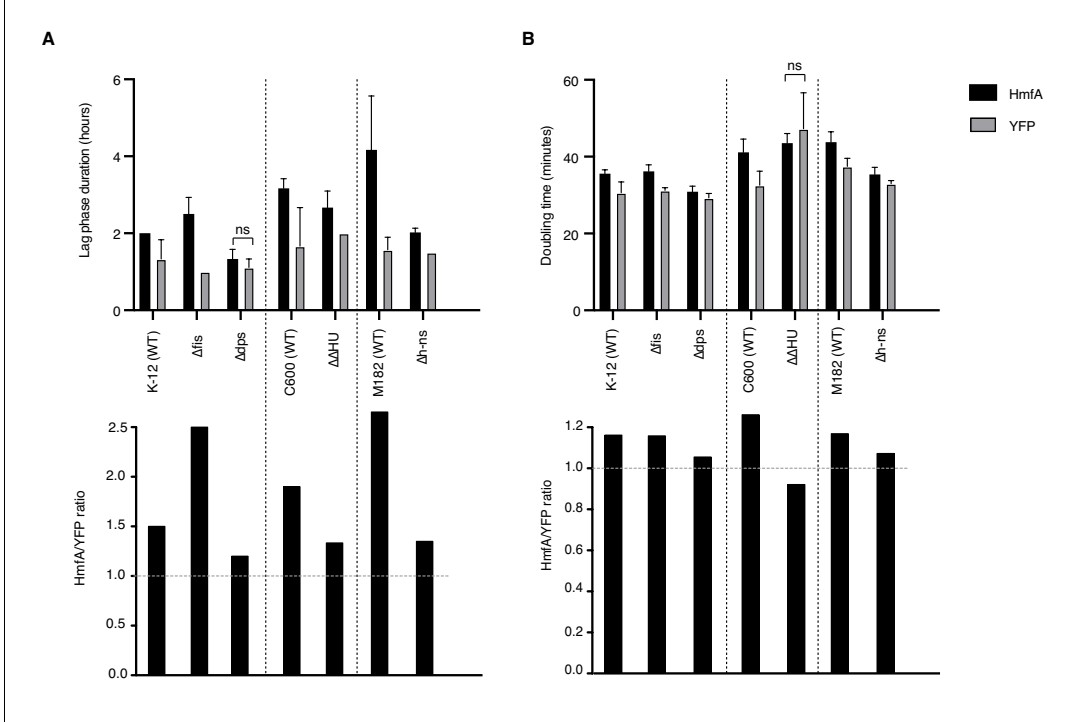

**Figure 8.** Effects of histone expression in NAP deletion strains. (**A**) Duration of lag phase, operationally defined as the time between inoculation and the point of maximum growth rate, in cultures of YFP- and HmfA-expressing cells from different genetic backgrounds. (**B**) Doubling time at the point of maximum growth rate as determined by curve fitting. All pairwise comparisons significant at p<0.05 unless indicated. Different genetic backgrounds are separated by dotted vertical lines.

DOI: https://doi.org/10.7554/eLife.49038.018

## Discussion

Our experiments demonstrate that archaeal histones are surprisingly well tolerated when expressed in *E. coli*, a system that has not evolved to deal with nucleosomal structures. Despite binding ubiquitously to the *E. coli* genome, they do not fundamentally compromise critical DNA-templated processes under favourable growth conditions. In particular, while we find some evidence that nucleosome occupancy locally restricts the output of the transcription machinery and that histones displace endogenous NAPs, gene expression is insufficiently perturbed to affect growth beyond a mild extension of lag phase. Transcriptomic analysis revealed induction of several stress responses as well as downregulation of DNA gyrases, which likely help the cells to adapt to the unique challenge of nucleosome formation. With the system already stretched, histones constitute a more severe problem when cells are forced to deal with double strand breaks or topological stress.

*E. coli* has not evolved to specifically deal with nucleosomal structures. Why then, did histone expression not cause much more drastic effects? We suggest that, both in *E. coli* and during evolution, global wrapping of DNA into nucleosomes was facilitated by two factors in particular: first, by virtue of their AT-rich nature, promoters remain comparatively accessible to the transcription machinery, even in a naïve prokaryote whose sequence and functional repertoire did not co-evolve to accommodate histones. Nucleosome-free regions at the TSS, a key features of nucleosome architecture in eukaryotes, might therefore have emerged, in the first instance, as a simple consequence of promoter composition. Once established, nucleosomes bordering the TSS were uniquely positioned to be co-opted into gene regulatory roles in eukaryotes and perhaps along different archaeal lineages, with nucleosome positioning later refined by evolution at specific loci to provide more nuanced control over transcriptional processes. Second, compared to their eukaryotic counterparts, archaeal nucleosomes appear to be more surmountable barriers to transcription elongation. Even at high histone concentrations, transcription through a HMf-chromatinized template in vitro is slowed but not aborted (*Xie and Reeve, 2004*), in line with the absence of recognizable histone remodelers

from archaeal genomes. Thus, near-global coating of the genome with archaeal-type histone proteins might have evolved without severe repercussions for basic genome function before a more restrictive arrangement, perhaps coincident with the advent of octameric histone architecture, took hold during eukaryogenesis. From an evolutionary point of view, one might therefore call the ground state mediated by archaeal histones proto-restrictive.

To what extent restrictive, proto-restrictive, or permissive ground states exist in different archaea in vivo remains unclear. Experiments with histones from *M. fervidus*, *Methanococcus jannaschii*, and *Pyrococcus furiosus* have shown that archaeal nucleosomes can interfere with transcription initiation and elongation in vitro (*Wilkinson et al., 2010*; *Soares et al., 1998*; *Xie and Reeve, 2004*; *Sanders et al., 2019*). However, significant inhibitory effects were only observed at high histone:DNA ratios (close to or above 1:1). Ratios of that magnitude, while regularly found in eukaryotes, need not be prevalent in archaea. Direct measurements of histone:DNA ratios are scarce and variable, with prior estimates in *M. fervidus* reporting stoichiometries as high as 1:1 (*Pereira et al., 1997*) and as low as 0.2–0.3:1 (*Stroup and Reeve, 1992*). Considering transcript levels as a (really rather imperfect) proxy, histones appear very abundant in *Thermococcus kodakarensis* and *Methanobrevibacter smithii* (*Figure 9*), strengthening the case for histones as global packaging agents in these species. In contrast, histone mRNAs are much less plentiful in *Haloferax volcanii* and *Halobacterium salinarum* (*Figure 9*), where histones likely have a limited role in DNA compaction (*Dulmage et al., 2015*) and less than 40% of the chromosome is resistant to MNase digestion (*Takayanagi et al., 1992*). In these species, non-histone proteins might be more important mediators of chromatin architecture and packaging. Thus, histone:DNA stoichiometry likely varies substantially across taxa as well as along the growth cycle (*Takayanagi et al., 1992*; *Dinger et al., 2000*; *Sandman et al., 1994*).

Attempts to delete histone genes have also revealed considerable diversity across archaea. Histones are required for viability in *T. kodakarensis* and *Methanococcus voltae* (*Čuboňováa et al., 2012*; *Heinicke et al., 2004*), but can be removed with surprisingly muted effects on transcription in *Methanosarcina mazei* (*Weidenbach et al., 2008*) and *H. salinarum* (*Dulmage et al., 2015*). In both species, a comparatively small number of transcription units were affected by histone deletion, the majority of which was down- rather than upregulated.

Taken together, these observations suggest that histones likely play a more variable, species- and context-dependent role in archaea, may only sometimes act as global repressive agents and, more

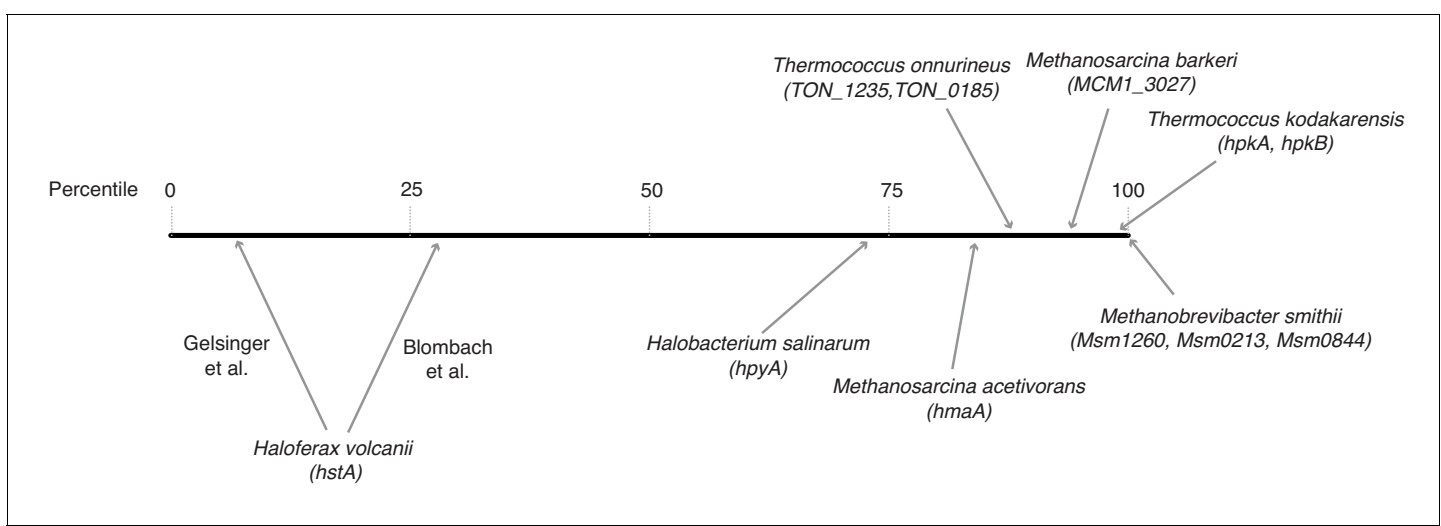

**Figure 9.** Relative transcript levels of histone genes across different archaeal species. Histones were assigned a percentile rank based on their relative expression in a given species and transcriptomic dataset (0 = least abundant mRNA in the dataset; 100 = most abundant mRNA in the dataset). For species with more than one histone gene, transcript levels were summed across histone genes. Because of significant variability between studies, two separate estimates are given for *H. volcanii*. Data sources: *H. salinarum* (Gene Expression Omnibus accession GSE99730), M. barkeri (GSE70370), *T. onnurineus* (GSE85760), *M. acetivorans* (GSE64349), *M. smithii* (GSE25408), *H. volcanii* (*Blombach et al., 2018* Nucl Acid Res 46:2308–2320; *Gelsinger and DiRuggiero, 2018* J Bacteriol 200:e00779-17), *T. kodakarensis* (*Jäger et al., 2014* BMC Genomics 15:684).
DOI: https://doi.org/10.7554/eLife.49038.019

generally, that care should be taken in projecting properties of eukaryotic histones onto those of archaea. In many instances, archaeal histones might be better understood with reference to bacterial NAPs, especially when considering how concentration drives opportunities for oligomerization, cooperativity, and bridging interactions with DNA. In this context, we note that our results are reminiscent of a recent study by Janissen and colleagues, who found that *dps* deletion in *E. coli* results in nucleoid decompaction but does not greatly impact transcription (*Janissen et al., 2018*). This provides some generality to the notion that architectural DNA-binding proteins, even if they bind to most of the genome and alter its compaction and gross structure, need not unduly interfere with transcription. The same study also highlights that, while polymerases may continue to access DNA and operate as usual, the same need not be true for other DNA-binding proteins: Dps substantially reduced the ability of several restriction enzymes to recognize and cut their target sites. Whether archaeal histones have similar effects in *E. coli* (beyond their ability to protect from MNase treatment), remains to be established. However, access regulation outside of a transcriptional context might well have provided the original evolutionary impetus for histones to spread across the genome, as genomes evolved to defend themselves against selfish elements that target the host genome for integration (*Talbert et al., 2019*). We note in this regard that our chromatinized *E. coli* strains might be of use for future synthetic biology applications. As more complex, combinatorial control of gene expression becomes a desirable genome engineering objective, limiting access to desired target sites will become an increasingly important design consideration (*Cardinale and Arkin, 2012*), as will chassis integrity in the face of potential invaders. As we find interference with transcription and replication to be limited, it will be interesting to experiment with expressing archaeal histones to restrict global access to the genome for specific DNA-binding factors or protect the genome against selfish element invasion (*Sultana et al., 2019*; *Aslankoohi et al., 2012*).

## Materials and methods

### Plasmid design

The coding sequences of *hmfA* and *hmfB* were codon-optimised for *E. coli* and synthesised as part of a rhamnose-inducible pD861 plasmid (*Figure 1—figure supplement 1*) by ATUM (Newark, CA). Originally, both plasmids also encoded a chromogenic protein to enable visual screening for induction. However, as the chromogenic protein was expressed at very high levels (*Figure 1—figure supplement 2*) and since we did not want to unduly increase cellular burden we removed the corresponding gene to yield pD861-hmfA. To generate non-binding histone mutants, *hmfA/hmfB* sequences were re-coded to carry three changes (K13T-R19S-T54K), previously shown to jointly abolish DNA binding of HMfB (*Soares et al., 2000*). These sequences were codon-optimized, synthesised and integrated onto a pD861 plasmid as above, without the chromogenic proteins, as was *hmfB*, for which cloning had failed. Plasmids pD861-hmfA, pD861-hmfB, pD861-hmfA$_{nb}$, and pD861-hmfB$_{nb}$ are identical expect for the sequences of the respective histone genes. *hmfA* was removed from pD681-hmfA to obtain Ec-EV.

### Bacterial transformation and growth

*E. coli* K-12 MG1655 cells were transformed via heat-shock with either pD861-hmfA, pD861-hmfB or pD861-EV, or the non-binding histone mutants pD861-hmfA$_{nb}$ or pD861-hmfB$_{nb}$ to generate strains Ec-EV, Ec-hmfA, Ec-hmfB, Ec-hmfA$_{nb}$ and Ec-hmfB$_{nb}$, respectively. All strains were grown in LB medium plus kanamycin (50 µg/ml) at 37°C with agitation (170 rpm). Histone expression was induced by adding L-Rhamnose monohydrate to a final concentration of 15 mM at OD600 ~ 0.6. Cells were harvested after 2 hr or 16–17 hr following induction.

### Protein purification

HMf protein purification was performed as in *Starich et al. (1996)*.

### Coomassie staining

Bacteria were harvested by centrifugation (4000 rpm for 15 min at 4°C), the supernatant discarded, and the pellet resuspended in a small volume of Histone Wash Buffer (50 mM Tris-HCl pH 7.5, 100 mM NaCl, 1 mM EDTA). Cell envelopes were disrupted using a Bioruptor Plus sonication system

(Diagenode s.a., Belgium) for 10 cycles, 30 s on/off with power set to high. The soluble protein fraction was separated from cellular debris by centrifugation at 15,000 x g for 15 min at 4°C, while the insoluble fraction was obtained by re-suspending the pelleted debris in Histone Wash Buffer. The protein concentration in the cell lysate was quantified with a Pierce BCA Protein Assay Kit (Thermo-Fisher Scientific, UK) using the provided albumin as standard. Protein fractions were separated by means of 16.5% Tris-tricine precast gels (Bio-Rad Laboratories, California) and bands were revealed by colloidal Coomassie (InstantBlue, Sigma-Aldrich) staining. Histone-expressing strains showed a band close to the size expected for HMfA/B (*Figure 1—figure supplement 2*). This band was excised and protein identity confirmed as HMfA/B via mass spectrometry.

## Growth assays

Overnight pre-cultures were diluted 1:500 into LB medium plus kanamycin (50 µg/ml). Samples were plated in replicate into a flat bottom Nunc 96-well plate (ThermoFisher Scientific, UK) and incubated at 37°C at 100 rpm for 30 min. OD measurement were performed using a high-throughput microplate reader (FLUOstar Omega, BMG LABTECH GmbH, Ortenberg, Germany) in which bacteria were grown at 37°C under continuous shaking (~500 rpm, double orbital). Optical density was measured at 600 nm every 30 min for 12.5 hr. For induction, the microplate reader was paused at cycle 6 and L-Rhamnose monohydrate added manually to the relevant wells to a final concentration of 15 mM. Results presented are from three biological replicates per strain, each averaged across six technical replicates.

## MNase digestion – *E. coli*

Bacterial cultures were harvested by centrifugation (4000 rpm for 15 min at 4°C), the supernatant discarded and the pelleted cells re-suspended in chilled 1x PBS (Gibco, ThermoFisher Scientific, UK). Cells were then fixed by adding a fixation solution (100 mM NaCl, 50mMTris-HCl pH 8.0, 10% formaldehyde) for 10 min at room temperature under slow rotation, after which fixation was quenched by adding 140 mM glycine. Following a further round of centrifugation (4000 rpm for 5 min at 4°C), bacteria were washed twice with 10 ml chilled 1x PBS and incubated in a lysozyme buffer (120 mM Tris-HCl pH 8.0, 50 mM EDTA, 4 mg/ml Lysozyme) for 10 min at 37°C to generate protoplasts. Cells were pelleted (15000 rpm for 3 min at room temperature) and re-suspended in 500 µl of lysis buffer (10 mM NaCl, 10 mM Tris-HCl pH 7.4, 3 mM MgCl2, 0.5% NP-40, 1x Pi, 0.15 mM Spermine, 0.5 mM Spermidine), transferred to a new microcentrifuge tube and incubated on ice for 20 min. Subsequently, the lysate was spun down and the pellet washed with 500 µl of -CA buffer (15 mM NaCl, 10 mM Tris-HCl pH 7.4, 60 mM KCl, 1x Pi, 0.15 mM Spermine, 0.5 mM Spermidine) without re-suspending. The washed pellet was finally re-suspended in 500 µl of +CA buffer (15 mM NaCl, 10 mM Tris-HCl pH 7.4, 60 mM KCl, 1 mM CaCl2, 0.15 mM Spermine, 0.5 mM Spermidine) to a uniform suspension. 50 µl of this suspension were digested with micrococcal nuclease (LS004798, Worthington Biochemical Corporation, NJ; 500 U/ml for Ec-hmfA and Ec-hmfB, 50 U/ml for Ec-EV) for 10 min (20 min for cells in stationary phase) at room temperature and finally blocked with a STOP solution containing calcium-chelating agents (100 mM EDTA, 10 mM EGTA). Each sample was further diluted with -CA buffer and treated with 10% SDS and 150 ng/ml proteinase K overnight at 65°C with shaking at 500 rpm. Undigested DNA fragments were purified by two rounds of phenol:chloroform extraction separated by an RNase A digestion step (100 µg/ml, 2 hr at 37°C with shaking at 500 rpm). Finally, DNA fragments were precipitated in ethanol and re-suspended in 40 µl distilled water. The quality of the digest and the size of the retrieved fragments were assessed by agarose DNA electrophoresis (2.5% agarose gel in 1x TBE run at 150V for 30 min).

## MNase digestion – *M. fervidus*

Frozen pellets of *M. fervidus* harvested in late exponential and stationary phase were purchased from the Archaeenzentrum in Regensburg, Germany. We then followed the MNase protocol outlined above with the following modifications: first, ~0.5 g of frozen pellet were thawed and re-suspended in 9 ml of 1x PBS before fixation. Second, due to differences in cell wall composition between *M. fervidus* and *E. coli*, the lysozyme digestion step was replaced by mechanical disruption with a French press: after the wash that follows fixation, the pellet was re-suspended in 20 ml of chilled 1x PBS, the cell suspension passaged twice through a TS Series French press (Constant Systems) at 15kpsi and then spun down at 4000 rpm for 15 min at 4°C before proceeding with cell lysis. Finally, the

extracted chromatin was re-suspended in 250 µl of +CA buffer (instead of 500 µl). Digestion, fragment purification, sequencing and analysis were performed as for *E.coli* but with a micrococcal nuclease concentration of 100 U/ml.

## MNase digest sequencing

Size distributions of the DNA fragments retrieved by MNase digestion of strains Ec-EV, Ec-hmfA, Ec-hmfB and *M. fervidus* were analysed with an Agilent Bioanalyser DNA1000 chip. For each of these strains, three biological replicates were selected for sequencing. Twenty nanograms per sample were used for library construction with the NEBNext Ultra II DNA Library Prep Kit for Illumina and NEBNext Multiplex Oligos for Illumina. The output was then taken to 10 PCR cycles and purified using a 1.8x Ampure XP bead clean-up kit. Libraries were quantified via Qubit and quality assessment carried out on an Agilent Bioanalyser DNA 1000 chip. Libraries were then sequenced on an Illumina MiSeq sequencer using single-end 160 bp reads.

## Read processing

Reads were trimmed using Trimmomatic-0.35 (single-end mode, ILLUMINACLIP:2:30:10) to remove adapter sequences. This did not remove short remnant adapter sequences so that we submitted reads to a further round of trimming using Trimgalore v0.4.1 with default parameters. Trimmed reads were aligned, as appropriate, to either the *Escherichia coli* K-12 MG1655 genome (NC_000913.3) or the *M. fervidus* DSM2088 genome (NC_014658.1) using Bowtie2 (*Langmead and Salzberg, 2012*). Only uniquely mapping reads were retained for further analysis. Per-base coverage statistics were computed using the genomeCoverageBed function in the bedtools2 suite (*Quinlan and Hall, 2010*).

## Peak calling

Nucleosome peaks were called using the NucleR package in R as described previously (*Hocher et al., 2019*). See *Supplementary file 2* for the relevant Fourier parameters.

## LASSO modeling

LASSO modeling was carried out for different footprint size classes ($60 \pm 5$ bp, $90 \pm 5$ bp, $120 \pm 5$ bp) using empty vector-normalized coverage. Empty vector coverage was computed across fragment sizes and coverage across the genome uniformly increased by one to enable analysis of zero-coverage regions. K-mer counts ($k=\{1,2,3,4\}$) were computed using the R seqTools package over windows of three different sizes (61 bp, 91 bp, 121 bp). Subsequent LASSO modeling was then carried out as described previously (*Hocher et al., 2019*), with models trained on one sixth of the *E.coli* genome (genomic positions 0–773608) and tested on the remainder of the genome.

## Transcriptional start sites

Experimentally defined transcriptional start sites were obtained from RegulonDB (*Salgado et al., 2013*) (http://regulondb.ccg.unam.mx/menu/download/datasets/files/High_throughput_transcription_initiation_mapping_with_5_tri_or_monophosphate_enrichment_v3.0.txt). The position inside each broad TSS associated with the most reads (column three in the file above) was defined as the TSS for downstream analysis. Promoter annotations were obtained from the same source (http://regulondb.ccg.unam.mx/menu/download/datasets/files/PromoterSet.txt).

## Comparison with other transcriptomes

All available transcriptomic data corresponding to *E.coli* K-12 strains were downloaded from the *E. coli* Gene Expression Database (GenExpdb, https://genexpdb.okstate.edu), which aggregates differential transcriptional responses (increased/decreased mRNA expression computed from pairwise comparisons in different individual studies). Similarity between differential expression in Ec-hmfA versus Ec-hmfA$_{nb}$ and other pairwise comparisons (*Figure 7A*) was calculated as the dot product of the two differential expression vectors.

## RNA extraction and sequencing

250 µl of culture were harvested from late exponential and stationary phase by centrifugation (15,000 x g at 4℃ for 15 min). The supernatant was discarded and the pellet re-suspended in 100 µl

of Y1 Buffer (1M Sorbitol, 0.1M EDTA, 1 mg/ml lysozyme, 0.1% β-mercaptoethanol) and incubated at 37°C for 1 hr at 500 rpm. The cell suspension was added to 350 µl of RLT buffer, 250 µl 100% ethanol and loaded onto an RNeasy column from the RNeasy Kit (Qiagen, Germany). RNA was then washed and eluted following the manufacturer's protocol. Eluted samples were incubated with DNase I (New England Biolabs, MA) for 10 min at 37°C and then cleaned up with a second passage through the RNeasy column (loading, washes and elution according to manufacturer's instructions). Samples were finally eluted in 30 µl of RNase-free water and RNA quantified with Nanodrop. Quality assessment of the extracted RNA was carried out with an Agilent Bioanalyser RNAnano chip and five replicates per strain/condition were chosen for sequencing.

## RNA sequencing

For each replicate/strain/condition, 1.5 µg of total RNA were depleted of rRNA using the Ribo-Zero rRNA depletion kit (Illumina) and libraries constructed using a TrueSeq Stranded RNA LT Kit (Illumina). After 12 PCR cycles, library quality was assessed with an Agilent Bioanalyser HS-DNA chip and quantified by Qubit. No size selection was carried out and the samples were sequenced on a HiSeq 2500 machine using paired-end 100 bp reads.

## Transcriptome analysis

Using Bowtie2, reads were first aligned to all annotated non-coding RNA genes (rRNA, tRNA, etc.). Reads that mapped to any of these genes were discarded, even if they mapped to more than one location in the genome. We then used Trim Galore v0.4.1 with default parameters to trim adapters and low-quality terminal sequences. Trimmed reads were aligned to the *E. coli* K-12 MG1655 genome (NC_000913.3) with Bowtie2 (–no-discordant –no-mixed). As a technical aside, we note that, despite the above filtering steps, some of the samples had an unusually low alignment rate (<30%). We found that most of the unaligned reads were perfect matches to rRNA sequences from *Bacillus subtilis* but not *E. coli* and had therefore eluded the above filter. As contamination at this scale is unlikely (no bacteria other than *E. coli* are grown or sequenced in the lab and a plain LB control was added to check for contamination when growing the samples for RNA extraction), we suspect these reads are the result of carrying over RiboZero oligos. The addition of a further round of filtering to discard reads that match non-coding RNA sequences from *Bacillus subtilis* increased the alignment rate to *E. coli* index up to ~90%.

By-gene read counts were computed from read alignments using the summarizeOverlaps function (mode='Union', singleEnd = FALSE, ignore.strand = FALSE, fragments = TRUE) from the GenomicAlignments package in BioConductor. Differential gene expression analysis was carried out using DESeq2 (*Love et al., 2014*). Replicates found to be outliers in principal component analysis and that were subsequently excluded from differential expression analysis are listed in *Supplementary file 3*.

## Microscopy

Overnight pre-cultures of Ec-EV, Ec-hmfA, and Ec-hmfB were diluted in fresh LB medium plus antibiotic and grown as described above. ~300 µl of culture were harvested by centrifugation (15,000 rpm for 15 min). Pellets were resuspended in 1% FA in PBS and fixed for 10 min at room temperature. Fixating agent was removed by spinning (15,000 rpm for 15 min) and pellets were resuspended in 1 ml PBS. 5 µl of cellular suspension was spread onto an agarose pad, covered in VectaShield containing DAPI (Vector Laboratories) and the excess liquid removed. Slides were imaged using a Manual Leica DMRB with phase contrast and DIC for transmitted light illumination. For quantification, images from three independent experiments were analysed with MicrobeJ (*Ducret et al., 2016*) to perform automatic cell detection and size measurements. MicrobeJ image profiles were manually curated to remove background and wrongly detected debris. For each sample/condition, measurements of cell length and area are derived from averages across ~10 independent pictures. Average DAPI profiles and associated cell contours were determined and visualized using Coli-Inspector (https://sils.fnwi.uva.nl/bcb/objectj/examples/Coli-Inspector/Coli-Inspector-MD/coli-inspector.html).

## NAPs binding regions

Genomic regions bound by Fis and H-NS were obtained from *Kahramanoglou et al. (2011)*, regions bound by IHF from *Prieto et al. (2012)*, and regions bound by Dps from *Antipov et al. (2017)*.

Differential histone occupancy was computed between regions bound by a given NAP and the unbound region immediately downstream.

## Modeling of HmfB oligomer stability and DNA affinity

The HMfB dodecamer unit cell was reconstructed from the crystal structure (PDB ID: 5t5k). Dimers were removed sequentially from the structure to build different sized HMfB polymers. We used the AMBER ff14SB forcefield (*Maier et al., 2015*) and solvated the unit cell with 14 Å of explicit TIP3P water and NaCl. We relaxed the system with 10,000 steps of steepest descent and conjugate gradient energy minimisation, heated the system to 300 K and performed 100 ns of NPT classical molecular dynamics using pmemd with a two fs timestep (*Salomon-Ferrer et al., 2013*). Binding affinity and stability calculations were performed using MMPBSA (*Miller et al., 2012*).

## Stress assays and NAP deletion mutants

To minimise the risk of suppressor mutations all cultures involving NAP deletions were inoculated straight from −80°C stocks. 5 mL LB cultures containing the antibiotic necessary for plasmid selection were inoculated in 50 mL tubes at OD = 0.1 from overnight LB cultures that contained antibiotics for both plasmid and mutant selection. After 2 hr, rhamnose was added to a final concentration of 15 mM to induce histone or YFP expression. Two hours after induction, 10 µL of cultures were used to inoculate 200 µL wells in Nunc 96-well microplates (at least three biological replicates, each with three technical replicates). Growth was monitored as described above. For chemical stress assays, novobiocin, rifampicin and $H_2O_2$ were used at respective concentrations of 10 µg/mL, 0.31 ng/mL, and 1.25 mM. For UV stress, cells used as inoculum were grown as for chemical stress assay and $10^6$ and $10^3$ dilutions (in 1x PBS) were plated onto plates containing 15 mM rhamnose and kanamycin. Plates were irradiated with 10000µJ using Stratalinker 2400.

## Data availability

Datasets generated for this study have been deposited in the NCBI Gene Expression Omnibus under accession number GSE127680 (https://www.ncbi.nlm.nih.gov/geo/query/acc.cgi?acc=GSE127680).

## Acknowledgements

We thank Ziwei Liang, Till Bartke, Ben Foster, and Kathleen Sandman for experimental advice, training, and sharing protocols; Finn Werner for his continued support and mentorship; Madan Babu, Ben Lehner, Peter Sarkies and members of the LMS Quantitative Biology section for discussions; Jacob Swadling for help with structure visualizations and modelling, the MRC LMS Genomics and Proteomics facilities for sequencing and mass spectrometry, and David Grainger and Jacques Oberto for NAP deletion strains. This work was supported by Medical Research Council core funding to TW. This project made use of time on UK Tier 2 JADE granted via the UK High-End Computing Consortium for Biomolecular Simulation, HECBioSim, supported by the EPSRC (grant no. EP/R029407/1).

## Additional information

### Funding

| Funder | Grant reference number | Author |
|---|---|---|
| Medical Research Council | MC_A658_5TY40 | Tobias Warnecke |
| Engineering and Physical Sciences Research Council | EP/R029407/1 | Tobias Warnecke |

The funders had no role in study design, data collection and interpretation, or the decision to submit the work for publication.

### Author contributions

Maria Rojec, Data curation, Formal analysis, Validation, Investigation, Visualization, Methodology, Writing—review and editing; Antoine Hocher, Formal analysis, Investigation, Visualization,

Methodology, Writing—review and editing; Kathryn M Stevens, Formal analysis, Investigation; Matthias Merkenschlager, Resources, Supervision, Writing—review and editing; Tobias Warnecke, Conceptualization, Resources, Formal analysis, Supervision, Funding acquisition, Investigation, Visualization, Methodology, Writing—original draft, Writing—review and editing

**Author ORCIDs**
Matthias Merkenschlager  http://orcid.org/0000-0003-2889-3288
Tobias Warnecke  https://orcid.org/0000-0002-4936-5428

**Decision letter and Author response**
Decision letter https://doi.org/10.7554/eLife.49038.039
Author response https://doi.org/10.7554/eLife.49038.040

## Additional files

**Supplementary files**
• Supplementary file 1. *E. coli* K-12 MG1655-derived strains constructed for this study.
DOI: https://doi.org/10.7554/eLife.49038.020

• Supplementary file 2. Fourier filtering parameters.
DOI: https://doi.org/10.7554/eLife.49038.021

• Supplementary file 3. Outliers in differential expression analysis.
DOI: https://doi.org/10.7554/eLife.49038.022

• Transparent reporting form DOI: https://doi.org/10.7554/eLife.49038.023

**Data availability**
Sequencing data have been deposited in GEO under accession code GSE127680.

The following dataset was generated:

| Author(s) | Year | Dataset title | Dataset URL | Database and Identifier |
|---|---|---|---|---|
| Rojec M, Hocher A, Merkenschlager M, Warnecke T | 2019 | The role of archaeal histones in gene expression - a synthetic biology perspective | https://www.ncbi.nlm.nih.gov/geo/query/acc.cgi?acc=GSE127680 | NCBI Gene Expression Omnibus, GSE127680 |

The following previously published datasets were used:

| Author(s) | Year | Dataset title | Dataset URL | Database and Identifier |
|---|---|---|---|---|
| Dulmage KA, Darnell CL, Vreugdenhil A, Schmid AK | 2017 | RNA-seq on rRNA depleted libraries from exponentially growing Halobacterium salinarum NRC-1 strains Δura3 and Δhlx2 | https://www.ncbi.nlm.nih.gov/geo/query/acc.cgi?acc=GSE99730 | NCBI Gene Expression Omnibus, GSE99730 |
| López Muñoz MM, Schönheit P, Metcalf WW | 2015 | Transcriptomic profiles of M. barkeri Fusaro DSMZ804 and Pyr+ strains | https://www.ncbi.nlm.nih.gov/geo/query/acc.cgi?acc=GSE70370 | NCBI Gene Expression Omnibus, GSE70370 |
| Cho S, Kim M, Jeong Y, Lee B, Lee J, Kang SG, Cho B | 2017 | Genome-wide primary transcriptome landscape reveals the diversity of regulatory elements in archaeal genomes | https://www.ncbi.nlm.nih.gov/geo/query/acc.cgi?acc=GSE85760 | NCBI Gene Expression Omnibus, GSE85760 |
| Fu H, Kohler PR, Metcalf WW | 2015 | High-throughput RNA sequencing of methanosarcina grown on methylated sulfur compounds | https://www.ncbi.nlm.nih.gov/geo/query/acc.cgi?acc=GSE64349 | NCBI Gene Expression Omnibus, GSE64349 |
| Hansen E, Rey F | 2011 | The pan-genome of the dominant human gut-associated archaeon, Methanobrevibacter smithii | https://www.ncbi.nlm.nih.gov/geo/query/acc.cgi?acc=GSE25408 | NCBI Gene Expression Omnibus, GSE25408 |

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
