## [Decision Letter]

**Acceptance summary:**

We commend the authors for the detailed responses to the initial decision. The manuscript has been notably improved by these efforts. While the major findings of this paper have largely stayed the same, the authors have discovered new subtleties that strengthen this study. By growing the archaeal histone expressing *E. coli* strains in additional growth conditions, the authors show that the histones inhibit growth under topological stress. Furthermore, they find that archaeal histones induce expression changes similar to growth arrest and H-NS deletion strains. Finally, the authors have taken pains to carefully address other concerns such as global transcriptional effects, new findings related to archaeal histones, and volumetric analyses of the cells. Thank you for submitting this fascinating study.

**Decision letter after peer review:**

Thank you for submitting your article "Chromatinization of *Escherichia coli* with archaeal histones" for consideration by *eLife*. Your article has been reviewed by three peer reviewers, one of whom is a member of our Board of Reviewing Editors, and the evaluation has been overseen by Jessica Tyler as the Senior Editor. The reviewers have opted to remain anonymous.

The reviewers have discussed the reviews with one another and the Reviewing Editor has drafted this decision to help you prepare a revised submission.

Summary:

Rojec and colleagues present a simple but rather provocative experiment: would expressing archaeal histones from *Methanothermus fervidus* in the histone-less bacterium *Escherichia coli* – a truly blank-slate for histones – recapitulate key features of chromatin-based regulation of transcription? The answer is both yes and no. Archaeal histone proteins have received renewed interest in recent years and important advances have been made in terms of understanding nucleosomal structures both biochemically and within cells. By expressing the archaeal histones hmfA and hmfB from the well-studied *M. fervidus*, many key characteristics about nucleosome binding were evident. The key findings are: (1) the histone proteins assemble genome-wide into multimeric structures of varying sizes according to known intrinsic sequence preferences; (2) nucleosome displacement during transcription and replication; (3) local transcriptional repression due to nucleosome binding at promoters and gene bodies; (4) modest to inconclusive global effects on transcription; and (5) surprisingly, no major repercussions to cell growth.

The manuscript is well-written and a pleasure to read. It reports a large body of convincing data on the neochromatinization of the *E. coli* genome with *M. fervidus* histones. The experiments have been planned with great care. A nice and novel observation is that the size of nucleosomes bound at transcription start sites correlates with repression efficiency. The paper will be intriguing for different subfields because it suggests that early histone-fold like proteins may not have evolved solely to regulate transcription, although the other posited roles remain hypothetical. Nevertheless, the amount of novel information generated in relation to archaeal histone function is limited: the reported results largely confirm earlier observations. We ask that the authors add more discussion as to whether there are additional biological insights about hmfA/B. Furthermore, the manuscript poses questions in the Introduction, which are perhaps, eukaryotic-centric. We ask that the authors reframe these questions as current work on archaeal histones suggests many similarities between bacteria and archaeal DNA organizing proteins. Finally, there was consensus that the authors will need to further justify their assertion that the effect to *E. coli* "…is remarkably unremarkable" and provide more analyses on the effects to transcription.

Essential revisions:

1) All reviewers agreed that the exponential growth experiments in rich medium (LB) were inadequate to support the statement that "growth of *E. coli* [with archaeal histones].… is remarkably unremarkable". Rich medium obviates the requirement for many *E. coli* genes for survival. Further, exponential growth has a relatively "fixed" transcriptome. In fact, the lagging phase (where transcriptional changes occur most) are likely where the nucleosomes have the largest effect. We would like to see a sampling of different types of stress: transcription (e.g., rifampicin), DNA damage (e.g., UV and/or chemical), metabolic (e.g., minimal media, oxidative stress, etc.). This can be done in a reasonable length of time. A more detailed analysis of gene expression, including the data they excluded, may offer clues as to the type of defect, if any, the histone expressing cells may be experiencing. Perhaps they could even include a cell cycle analysis.

2) It is rather striking that the hmfA/B expressing strains have both larger cell size and greater size heterogeneity. This was only briefly mentioned in the Results but is rather striking. The fact that the SOS response is activated suggests that the histones are affecting more processes than suggested. Would growing the cells in conditions that cause DNA stress (UV for example) exacerbate the morphology? Are the cells overall bigger or just narrower and longer (the cell area analysis seems to favor the former)? Could this be due to increased volume of chromatinized DNA? We would be satisfied with a more thorough discussion on this topic but would also like to see whether HfmA/B/nb expressing cells display additional morphological changes. It would also be useful if the authors could present additional information, if available, on cell-to-cell variability in terms of expression of the histone proteins. This might aid in making their point that hmfA/B have little impact on cell physiology.

3) We generally found that the "locally repressive effects" was not robust and needed further exploration. The authors could partially address this by explicitly showing, examining, and categorizing by function the genes that are down- or up-regulated in hmfA/B/nb vs. EV, and determine the corresponding level of nucleosome signal in hmfA/B expressing cells. They reasonably exclude these as potentially generic responses to ectopic protein expression. But is that the case? Do these genes change expression in response to any ectopically expressed proteins or are there residual hmfAnb effects? The idea is to deconvolute whether histones incorporate more readily with already downregulated genes or cause their repression. Could these include genes that enable *E. coli* to adapt to histones regardless of whether they are bound to DNA? Finally, the correlation between GC content vs. histone occupancy is not formally quantified (only browser tracks are shown). What is the relationship genome-wide? Is it significant? Here we are only asking for additional analyses of the existing data.

4) We ask for additional analyses on bacterial organizational structure in the context of AT/GC compared to endogenous bacterial NAPs. What is conceptually the difference between over-expressing a bacterial DNA organizing protein, one from another bacterium, or an archaeal histone? Why do we not see a phenotype similar to that of *E. coli* lacking one or more of its own DNA organizing proteins? We would like to see some straightforward estimates (using RT-qPCR) of a few stress-regulated genes under direct control of bacterial NAPs when grown under stress. We also ask whether there is phenotypic complementation of H-NS and HU knockout strains. In the case of HU this would make clear whether the similar structural properties of the two proteins are enough to have complementation. In the case of H-NS it could suggest that invasion of hmfA/B into large H-NS regulated regions indeed has only mild effects on global transcription. Such growth experiments carried out under a few conditions (see 1) should be fairly straightforward. What is the difference between AT-rich non-histone binding regions and AT-rich H-NS binding regions? What is the effect of histones in gene bodies; is some of the repression of spurious intragenic transcription by H-NS taken over by bound archaeal histones?

5) One of the most interesting questions put forward by the authors is 'how easy or hard is it to transition from a system without histones to one where histones are abundant?', but the answers to this question leaves something to be desired. Although the authors use an inducible system (2 h and 16 h), perhaps only subpopulations of the cells accommodate histones? We think it is important to rule out potential adaptive mechanisms especially by looking more deeply into the gene expression changes. It is quite possible that this occurs at the level of single-cells, which is beyond the scope of this study, but perhaps explains some of the variability. We again suggest simple experiments such as determining whether the SOS response is required to tolerate expression of histones or their binding to DNA.

---

## [Author Response]

[…] The amount of novel information generated in relation to archaeal histone function is limited: the reported results largely confirm earlier observations. We ask that the authors add more discussion as to whether there are additional biological insights about hmfA/B.

We share the reviewers’ desire to learn more about the native functions of archaeal histones. Given that we express archaeal histones in a heterologous system, we have been cautious in drawing firm conclusions about native functionality. That said, we do think that our experiments have revealed interesting leads that might help us better understand how the histones of *M. fervidus* and other archaea operate in their native cellular context. Even though not the focal point of the manuscript, it is worth highlighting two such leads:

First, as highlighted by the editor, we provide some evidence – yet to be reported for any archaeon – that longer oligomers elicit stronger repression. We now also provide evidence from molecular dynamics simulations that longer oligomeric complexes are more stable (Figure 4—figure supplement 3). It is certainly conceivable that archaea make use of this property to dynamically regulate genome accessibility, either by changing absolute histone concentrations (which should lead to longer oligomers on average – see exponential-stationary transition in *E. coli* in Figure 1C) or by varying the relative levels of different histone paralogs that are more or less likely to form stable longer oligomers.

Second, while our focus in this manuscript is on the question how well a naïve system tolerates histone proteins, we recently used some of the MNase data reported here to understand sequence-dependent oligomer formation in archaea themselves (Hocher et al.,2019 bioRxiv 564930). There, we suggest that oligomer formation is subject to sequence-dependent nucleation-extension dynamics. We show that initial tetramer formation relies on a relatively high affinity site but that, from this nucleation point, oligomers can reach into less conducive sequence space. Importantly, this might enable repression of promoters that – by virtue of their sequence composition – disfavour nucleation. We anticipate that our heterologous *E. coli* system, which displays similar nucleation-extension dynamics (see Author response image 1, Figure 2 —figure supplement 1 in the revised manuscript), will help to elucidate this further, especially since targeted manipulations (e.g. of promoter sequences) remain impossible or difficult for the majority of archaea. We now highlight these insights in the subsections “Intrinsic sequence preferences govern nucleosome formation along the *E. coli* genome and “Evidence that nucleosome formation locally represses transcription”.

**Author response image 1. respfig1:** The relationship between oligomer formation and the underlying sequence. (**A**) The upper half of this figure corresponds to Figure S4b in Hocher et al.2019, bioRxiv: 564930. The data are for *M. fervidus* native chromatin digests. The figure compares different types of narrow peaks, i.e. peaks principally formed by tetramers. Narrow peaks where longer fragments are common (i.e. extension is likely to occur, central panel) are compared with narrow peaks where longer fragments are rare (i.e. extension is uncommon, right panel). Narrow peaks where extension rarely happens are associated with significantly elevated flanking AT content (see Hocher et al. for quantification, W=A or T, S=G or C). (**B**) Patterns very similar to those observed in (**A**) are also evident in *E. coli* expressing HMfA, supporting the notion that sequence-dependent oligomerization behaviour is an intrinsic property of archaeal histones.

Regarding insights into HMfA versus HmfB in particular (rather than archaeal histones in general) we note that differences between the two paralogs clearly do exist, as evident from MNase profiles (Figure 1C, Figure 2E), the transcriptomic response (Figure 4, 7) and prior work by John Reeve’s group. However, these differences are comparatively subtle. As this issue is not central to the manuscript, we decided not to focus on these differences further. However, our data will allow a more focused exploration of this in the future.

Furthermore, the manuscript poses questions in the Introduction, which are perhaps, eukaryotic-centric. We ask that the authors reframe these questions as current work on archaeal histones suggests many similarities between bacteria and archaeal DNA organizing proteins.

We agree that there are important similarities between bacterial NAPs and archaeal histones. In fact, we stated this forcefully in the Discussion.

“… care should be taken in projecting properties of eukaryotic histones onto those of archaea. In many instances, archaeal histones might be better understood with reference to bacterial NAPs, especially when considering how concentration drives opportunities for oligomerization, cooperativity, and bridging interactions with DNA.”

It is therefore not because we are unaware of these similarities that we pose eukaryotic-centric questions in the Introduction. Rather, the framing is a considered choice. We believe that most readers will approach this manuscript from a eukaryote-based knowledge of histones and histone function. We aim to pick them up from their place of relative comfort and journey towards a better understanding of archaeal histones, ending up with a more nuanced understanding of the nature of archaeal histones – functionally somewhere between eukaryotic histones and bacterial nucleoid-associated proteins.

Finally, there was consensus that the authors will need to further justify their assertion that the effect to E. coli "…is remarkably unremarkable" and provide more analyses on the effects to transcription.

We appreciate that what is and is not remarkable is very much in the eye of the beholder. We have therefore characterized growth in a number of different stress conditions, as detailed below (point #5). We have also added a more rigorous comparative analysis of global and local transcriptional effects. To reflect that our astonishment is necessarily subjective we have rephrased the relevant sentence (subsection “Histone binding is associated with mild phenotypic effects under favourable conditions”). Whether the reviewers and readers share our astonishment is, of course, entirely up to them.

Essential revisions:1) All reviewers agreed that the exponential growth experiments in rich medium (LB) were inadequate to support the statement that "growth of E. coli [with archaeal histones].… is remarkably unremarkable". Rich medium obviates the requirement for many E. coli genes for survival. Further, exponential growth has a relatively "fixed" transcriptome. In fact, the lagging phase (where transcriptional changes occur most) are likely where the nucleosomes have the largest effect. We would like to see a sampling of different types of stress: transcription (e.g., rifampicin), DNA damage (e.g., UV and/or chemical), metabolic (e.g., minimal media, oxidative stress, etc.). This can be done in a reasonable length of time.

We share the reviewers’ curiosity to see how our histone-expressing strains respond to various stresses. In the revised manuscript we now present data on growth in response to rifampicin (transcriptional stress), H_2_O_2_ (oxidative stress), UV (DNA damage), and novobiocin treatment (supercoiling stress). We added the latter condition to explore whether histones, known to constrain negative supercoils, exacerbate the effects of gyrase inhibition. To stress-test the cells, we slightly modified the growth protocol, inducing histone expression in exponentially growing cells with rhamnose, growing these cells for a further two hours in rhamnose, and then using them to inoculate a new culture that contains rhamnose as well as the drug of interest (rifampicin etc). In this way, we ensure that stress responses are measured in cells where histones are already established, and that we can adequately capture effects of histone occupancy during lag phase. We then tracked growth/survival of these strains by measuring optical density or, in the case of UV irradiation, colony formation. The results of these experiments are displayed in Figure 6.

Briefly, the salient insights are as follows:

- When pre-induced cells are transferred into new LB medium, there is a slightly prolonged lag phase. However, histone-expressing strains recover quickly and eventually catch up with non-binding/EV control strains.

- Lag phase is extended further in strains treated with rifampicin or H_2_O_2_. Again, histone-expressing strains recover well. Under these conditions, histones can therefore be said to have a mild bacteriostatic but no bacteriocidal effect.

- In contrast, the presence of histones matters in the face of UV and novobiocin treatment, where colony formation and growth, respectively, are severely affected. In novobiocin-treated cells, we observe strong morphological changes, as cells become conspicuously elongated (Figure 6). The presence of histones therefore seems to interfere with the cells’ ability to effectively deal with double strand breaks (UV) and altered supercoiling levels.

Based on these results, which are complemented by new insights from transcriptomic analysis (see below), it would be reasonable to surmise that histones do not compromise dynamic responses to stress in general. However, their presence is problematic where the stress specifically relates to sensing or dealing with altered DNA topology or damage. Given what we know about histones, this is not unexpected. We present and discuss these results in the subsection “Systemic transcriptional responses to histone expression in *E. coli*”.

A more detailed analysis of gene expression, including the data they excluded, may offer clues as to the type of defect, if any, the histone expressing cells may be experiencing. Perhaps they could even include a cell cycle analysis.

We have carried out substantial further analysis of our gene expression data, including uncensored data, and now provide a more in-depth comparison of our strains to other perturbations that had previously been assessed via global transcriptome profiling (https://genexpdb.okstate.edu/). This allows us to place the effect of histone expression into a comparative framework and, by virtue of similarity to other conditions, suggest molecular pathways through which histones affect *E. coli* physiology.

Figure 7A summarizes the key results of this analysis. It displays the similarity in transcriptional responses between histone expression (Ec-hmfA versus Ec-hmfA_nb_) and previous experiments that involved differential gene expression analysis.

Comparing differential expression in exponential phase Ec-hmfA (versus Ec-hmfA_nb_) to other experiments, several interesting patterns emerge:

- The most similar differential expression patterns are observed for other histone-expressing strains, regardless of whether non-binding mutants or the empty vector control is used as a comparator. This provides further reassurance that both EV and the non-binding histone constitute reasonable controls.

- In terms of genes that are down-regulated, the histone-expressing strains exhibit global transcriptional similarities to amino acid starvation, cadmium shock, and heat stress. What unites these conditions is a transient growth arrest phenotype and induction of the stringent response as well as the general stress response (rpoS regulon). Genes in the rpoS regulon are also upregulated in histone-expressing strains (Figure 7B) while flagellar genes are downregulated, a response our strains share with the stringent response.

- In terms of genes that are up-regulated, histone-expressing strains are similar to conditions associated with environments where carbon sources are scarce/exhausted (stationary phase, minimal media) or altered (glucose-to-lactose shift).

These transcriptional signatures of transient growth arrest and/or slowed growth are in line with the growth phenotypes we observe, which are characterized by an extended lag period (see Figure 7B). This suggests that histone expression slows initial growth until cells have had time to adjust. Such adaptive adjustments almost certainly involve the downregulation of gyrases, as previously noted. Histones wrap DNA in negatively constrained supercoils. Gyrases introduce negative (or relax positive) supercoils so counteracting the histone-associated built-up of negative supercoiling by downregulating gyrases makes sense. Downregulation of gyrases then likely renders cells more susceptible to novobiocin (see Figure 7A). This model is further consistent with the observation that expressing CcdB, a gyrase poison, leads to gene expression changes that have some similarity to histone expression (Figure 7A).

We were further intrigued to see that, with regard to upregulated genes, the effect of histone expression is similar to the effects of deleting *h-ns*.

Finally, and importantly, we note that “similarity in expression” here is strictly relative. The strongest correlation to a dataset outside this study is modest (ρ=0.34). Thus, the transcriptional response to histone expression by no means precisely mimics any of the perturbations discussed above but has a strong unique component.

2) It is rather striking that the hmfA/B expressing strains have both larger cell size and greater size heterogeneity. This was only briefly mentioned in the Results but is rather striking. The fact that the SOS response is activated suggests that the histones are affecting more processes than suggested. Would growing the cells in conditions that cause DNA stress (UV for example) exacerbate the morphology?

We have considered how stress might exacerbate the impact of histone expression on cell morphology in the context of novobiocin treatment. Gyrase inhibition does indeed exacerbate morphology in histone-expressing cells, which become extremely elongated (see Figure 6).

Are the cells overall bigger or just narrower and longer (the cell area analysis seems to favor the former)? Could this be due to increased volume of chromatinized DNA? We would be satisfied with a more thorough discussion on this topic but would also like to see whether HfmA/B/nb expressing cells display additional morphological changes.

We agree that it is interesting to explore changes to nucleoid morphology and nucleoid/cytoplasm ratio in histone-expressing *E. coli* cells. As a potential starting point for future more in-depth analysis, we therefore decided to carry out a quantitative analysis of the relationship between nucleoid and cell volume based on DAPI-stained cells. As evident from Figure 5—figure supplement 1, we find no obvious changes to the nucleoid/cytoplasm ratio in histone-expressing cells and therefore no evidence for either compaction or decompaction. Ec-hmfA/B have a somewhat larger relative nucleoid volume than Ec-EV, but are indistinguishable from WT, suggesting that this is an Ec-EV-specific effect.

It would also be useful if the authors could present additional information, if available, on cell-to-cell variability in terms of expression of the histone proteins. This might aid in making their point that hmfA/B have little impact on cell physiology.

We initially considered tagging the histones with a fluorescent reporter, which would have allowed direct quantification. However, prior attempts at tagging by others were abandoned as even small tags were found to affect DNA binding (Sandman, pers. comm.). Tagging is much easier in eukaryotic histones, where the tag can be attached to the long flexible tails that are absent from archaeal histones. As an imperfect alternative, we sought to approximate heterogeneity by monitoring cell-to-cell variability of YFP expressed from the same promoter, either with or without HMfA simultaneously expressed (from a different plasmid but the same promoter) in the same cell. What we can conclude from Author response image 2 is somewhat limited but, to us, two features stand out: first, YFP is well-expressed in the majority of cells that also express HMfA. These estimates should be conservative given that, when HMfA is in play, both YFP and HMfA compete for transcriptional/translational resources. Second, there is an enrichment of cells with low YFP expression when HmfA is overexpressed. This is consistent with translational arrest in a subset of cells, in line with the transient growth arrest/stringent response phenotype we observe.

**Author response image 2. respfig2:** YFP expression measured in cells with or without concurrent expression of HmfA.

3) We generally found that the "locally repressive effects" was not robust and needed further exploration. The authors could partially address this by explicitly showing, examining, and categorizing by function the genes that are down- or up-regulated in hmfA/B/nb vs. EV, and determine the corresponding level of nucleosome signal in hmfA/B expressing cells. They reasonably exclude these as potentially generic responses to ectopic protein expression. But is that the case? Do these genes change expression in response to any ectopically expressed proteins or are there residual hmfAnb effects? The idea is to deconvolute whether histones incorporate more readily with already downregulated genes or cause their repression.

We think the reviewers make several distinct points here. Below, we have tried to disaggregate these points as we understand them.

a) The reviewers would like to see additional information to support the finding of locally repressive effects. They reasonably suggest that an alternative explanation might be true, namely that cause and effect are reversed, i.e. increased histone occupancy follows (rather than elicits) downregulation. There are two key pieces of evidence that strongly argue against this interpretation. First, as now highlighted in Figure 3F, unlike sometimes observed in eukaryotes, there is no negative global relationship between mRNA levels (as measured in Ec-EV to approximate the pre-histone state) and histone occupancy in our system. In fact, the relationship is slightly positive (ρ≈0.13-0.18, depending on histone type and growth phase). Thus, lower expression per se is not associated with higher histone occupancy. Second, we showed previously (Figure 4—figure supplement 2) that the relationship between histone occupancy and expression, while present at the promoter, breaks down when considering a control window just downstream of the promoter. This is inconsistent with the alternative model where downregulation precedes histone deposition. If histones simply accumulated on the DNA because transcription was reduced or switched off, the same relief from transcriptional perturbation should be observed immediately downstream of the promoter. It is not.

b) The reviewers also wonder, if we understand correctly, whether finding downregulation in the binding strain *and* the non-binding strain necessarily means that downregulation in the binding strain is really unrelated to DNA-binding. This is a good question. To begin with, we cannot and do not claim that coincident downregulation in both the binding and non-binding strain implies that downregulation in the binding strain is necessarily unrelated to binding. It is perfectly possible, for example, that histones exert a mild repressive effect at gene X and the same gene X in the non-binding histone strain also experiences downregulation, but for different reasons. We chose to focus on genes that are up- or down-regulated *only* in strains with a DNA-binding histones to minimise type I errors. Our overriding concern here is to isolate the effects of DNA-binding on gene expression. Naturally, in doing so we almost inevitably decrease sensitivity and miss genes at which histones exert a genuine effect via DNA binding and nucleosome formation. In the context of obtaining robust results on the direct effects of nucleosome formation, we believe this is a price worth paying.

c) Finally, it is possible, that non-binding mutant histones still associate with DNA. The unexpectedly poor growth upon novobiocin treatment also hints at that possibility for HmfA_nb_. However, we have shown experimentally that the non-binding mutants do not form nucleosomes and therefore would have to associate with DNA in a qualitatively distinct manner. Does that mean that overexpression of non-binding histones has impacts that are distinct from expressing other proteins? Yes. Is this a good thing? We would argue that it is, since it provides a more tailored control for other features of histones (their aggregation potential, propensity to interact with other peptides, etc.) that would not be controlled for by expressing a more inert gratuitous protein, and it is this well-matched control that allows us to home in on the specific effects of nucleosome formation along the *E. coli* chromosome.

We struggled to follow the reviewers’ comment that categorizing genes by function would shed light on the issues above. There is no a priori reason to suspect that direct, locally mediated repressive effects of histone occupancy are divided along functional lines (rather than, say, driven by sequence composition at the promoter). One would expect functional structure for transcriptional responses launched to cope with the stress induced by histones (in terms of both heterologous expression and stress associated with binding), but not in terms of the initial targets of local repression. NB: we did carry out a classic gene set enrichment analysis (GSEA), which showed depletion of genes in COG M (caused by flagellar genes) and COG J (translation), very much in line with the findings from comparative transcriptomic analysis. We did not present those results for two reasons: first, we think the comparative transcriptomic analysis is more insightful. Second, we think that toolkit to carry out GSEA in prokaryotes is fundamentally incomplete. Tools inherited from eukaryotes do not take into account operons. This is problematic because genes cluster into operons by function. As a consequence, changes to the regulation of a single operon get amplified by the number of genes of the same function that operon contains. In light of this problem, it is typically unclear what statistical enrichments in prokaryotic GSEA actually mean.

Could these include genes that enable E. coli to adapt to histones regardless of whether they are bound to DNA?

Yes, absolutely, although it is arguably harder to pinpoint specific genes that react in this manner (i.e. respond specifically to histones rather than any heterologous protein, but respond irrespective of binding). Our experiments were not designed to address this specifically.

Finally, the correlation between GC content vs. histone occupancy is not formally quantified (only browser tracks are shown). What is the relationship genome-wide? Is it significant? Here we are only asking for additional analyses of the existing data.

We already provided a quantitative assessment of the relationship between GC content and histone occupancy in Figure 3, in particular 3D. There are strong correlations in stationary phase between GC content and histone occupancy (normalized by coverage in the empty vector strain to control for MNase-associated biases; ρ=0.64 for hmfA, P<2.2x10^-16^).

4) We ask for additional analyses on bacterial organizational structure in the context of AT/GC compared to endogenous bacterial NAPs. What is conceptually the difference between over-expressing a bacterial DNA organizing protein, one from another bacterium, or an archaeal histone? Why do we not see a phenotype similar to that of E. coli lacking one or more of its own DNA organizing proteins? We would like to see some straightforward estimates (using RT-qPCR) of a few stress-regulated genes under direct control of bacterial NAPs when grown under stress. We also ask whether there is phenotypic complementation of H-NS and HU knockout strains. In the case of HU this would make clear whether the similar structural properties of the two proteins are enough to have complementation. In the case of H-NS it could suggest that invasion of hmfA/B into large H-NS regulated regions indeed has only mild effects on global transcription. Such growth experiments carried out under a few conditions (see 1) should be fairly straightforward. What is the difference between AT-rich non-histone binding regions and AT-rich H-NS binding regions? What is the effect of histones in gene bodies; is some of the repression of spurious intragenic transcription by H-NS taken over by bound archaeal histones?

We agree with the reviewers that it would be interesting to understand in greater molecular detail how histone expression impacts nucleoid organization, how histones interact with endogenous NAPs, and whether histone expression poses a challenge distinct from overexpressing one of these NAPs.

We now present several new lines of investigation that shed light on these questions. We identified intriguing similarities in the global transcriptional response between hmfA/B-expressing strains and an *h-ns* deletion strain (ρ=0.19, P<2.2x10^-16^). As direct targets are arguably better characterized for H-NS than any other NAP, we focused our investigation on H-NS. We began by comparing gene expression changes in our strain to those in a previously published *h-ns* deletion strain (same K-12 background, Gawade et al., 2019). Perhaps most notably, genes previously identified as direct H-NS targets (green rectangles in Figure 7C), are amongst the most upregulated genes not only when *h-ns* is deleted (as one would expect), but also upon HMf expression. This might indicate, as suggested by the reviewers, that histones displace H-NS, but fail to provide similar silencing, leading to de-repression of H-NS target genes. This hypothesis is consistent with our previous analysis (now Figure 7D) showing that histones are not significantly excluded from previously determined H-NS binding sites, implying that they successfully compete for binding at those sites.

We also attempted to look at spurious (intragenic and intergenic) transcription in this dataset. However, we found that the proportions of reads mapping to intergenes or antisense were effectively indistinguishable in the *h-ns* deletion and the corresponding wildtype strain. In other words, we cannot detect what are considered typical effects of H-NS removal in this dataset. This might be because spurious transcripts are short and/or unstable and therefore not well represented in data from a typical RNA-Seq workflow (compared to TSS mapping as done, for example, by David Grainger’s lab). Irrespective of the underlying reason, this precludes an informed comparison of spurious transcription between this and our data. As the focus of this manuscript is not on H-NS, we did not pursue this further experimentally.

In contrast to Δ*h-ns*, we found no similarities of note when comparing our data to other NAP deletion mutants (Δ*hupA/hupB*, Δ*dps,* Δ*fis,* all ρ<|0.04|). As far as we are aware, transcriptome-wide data on the effects of NAP overexpression (vis-à-vis deletion) in a comparable genetic background are not available. They would also be intrinsically harder to compare and interpret given that a) effects will inexorably depend on the level to which the protein is overexpressed and b) phenotypes reflect expression burden as well as protein-specific effects unless suitable controls are in place. However, it is interesting to note that strong (>40-fold) overproduction of H-NS has previously been reported to lead to a transient (several-hour) growth arrest after which cells resume growth (McGovern et al.,1994). This situation, which the authors dubbed “artificial stationary phase”, is qualitatively reminiscent of the prolonged lag phase we observe in the presence of HMf (see point #5) and further underlines the transcriptional similarity between histone-expressing strains and *E. coli* growing in stationary-phase(-like) conditions. Why would histone expression resemble both *h-ns* deletion and *h-ns* overexpression? We suggest, based on the chain of events proposed above, that histone displacement of H-NS not only leads to dysregulation of target genes, but – like overexpression – also generates a pool of free H-NS that might go on to bind off-targets (including perhaps AT-rich promoters) or have toxic effects unrelated to DNA binding.

In contrast to H-NS, HU overexpression from the same promoter led to 1.8-fold slower growth but no growth arrest, again underlining the outlier status of H-NS.

Our results, both old and new, indicate that histones likely invade genomic real estate normally occupied by NAPs. Histones also share some key properties with NAPs. Notably, they resemble HU in their ability to constrain negative supercoils. We therefore agree that it is interesting to ask if histones complement NAP deletion mutants. To address this question, we have assayed effects of HMfA expression on growth in a collection of NAP deletion strains. The results are summarized in Figure 8, considering differences in the time to maximum growth rate (an operational proxy for the duration of lag phase) and doubling time at the time of maximum growth as determined by curve fitting (see Materials and methods). We note the following:

- NAP deletions in *E. coli* are not associated with a strong growth phenotypes, with two exceptions:

*fis* deletion (Δ*fis*) is associated with a reduced lag phase;

the *hupA/hupB* deletion (ΔΔHU) strain grows more slowly (increased doubling time) compared to its C600 WT progenitor strain.

- HMfA expression generally leads to an increase in lag phase duration, which is particularly pronounced in Δ*fis* (and – for unknown reasons – in M182 WT)

- HMfA expression is associated with a relatively small but consistent increase in doubling time, but this effect does not appear to be compounded in NAP deletion strains. Interestingly, the growth retardation associated with *hupA/hupB* deletion and HMfA expression are not additive, suggesting that histone expression might partially alleviate defects associated with the absence of HU.

Note that we use YFP here for historical reasons – we had already carried out some of these assays early on in the project, prior to constructing the non-binding histone control strains. YFP serves as a control for the burden imposed by gratuitous expression.

Finally, we want to comment more generally on whether there are (what the reviewers call) “conceptual differences” between over-expressing a histone or an endogenous or heterologous bacterial NAP. We believe there are differences, but we would not call them conceptual. Rather, differences are grounded in the specific properties of histones vis-à-vis DNA binding proteins found in bacteria as well as in evolutionary context. What are these differences? To begin with, histones are arguably unusual in the way they wrap DNA and in the fact that they bind DNA strongly enough to provide protection from MNase. One might therefore suspect that they provide a stronger barrier than many other NAPs to the molecular machines that need to access DNA. In addition, many well-characterized bacterial NAPs, certainly those in *E. coli*, prefer binding to AT-rich DNA. This might be because they have an ancestral or extant function in the silencing of genomic invaders, which – for reasons that remain poorly understood – tend to have a GC content lower than the host genome. Histones, in contrast, prefer the GC-richer parts of the genome. This has some interesting implications. For example, as we suggest in the manuscript, histones might naturally have a lesser impact on sequences like promoters that are naturally AT-rich, reducing their impact on transcription. These divergent sequence preferences might also imply that, at low dosage, histones and NAPs such as H-NS might come to occupy different parts of the genome. If dosage is sufficiently high, however, histones, having saturated their preferred binding sites, will start to compete with NAPs for their preferred targets. Of course, these differences are not dogmatic. For example, we recently showed that a HU ortholog in *Thermoplasma acidophilum* prefers GC-rich sequences and behaves in many respects more like an archaeal histone than a “classic” bacterial HU (Hocher et al.,2019). Arguably the biggest difference, however, is evolutionary context: the absence of molecular machinery that has co-evolved with histone proteins.

5) One of the most interesting questions put forward by the authors is 'how easy or hard is it to transition from a system without histones to one where histones are abundant?', but the answers to this question leaves something to be desired. Although the authors use an inducible system (2 h and 16 h), perhaps only subpopulations of the cells accommodate histones?

The observation that, under standard conditions, histone expression has a mild bacteriostatic but no bacteriocidal effect argues strongly against the possibility that a significant fraction of cells drop out of the population because they cannot accommodate histones. That does not mean, of course, that there is no variability in histone expression and that some cells have an easier time than others, but note that across the population we observe histone:DNA ratios of up to ~0.7:1, so cells with a reduced burden are offset against those where the ratio exceeds 0.7:1.

We think it is important to rule out potential adaptive mechanisms especially by looking more deeply into the gene expression changes. It is quite possible that this occurs at the level of single-cells, which is beyond the scope of this study, but perhaps explains some of the variability. We again suggest simple experiments such as determining whether the SOS response is required to tolerate expression of histones or their binding to DNA.

To understand the role of the SOS response in allowing *E. coli* cells to adapt to histone expression, we assayed growth in a *recA* deletion strain expressing histones. Histone expression leads to a significant reduction in growth compared to the control but is by no means lethal (see Figure 7—figure supplement 1). This suggests that the SOS response facilitates adaptation to some aspect of the presence of histones, but is not critical for survival. Consistent with the SOS response being triggered following histone expression, we also show concerted upregulation of the rpoS regulon in our strains (see above).

On a more general note, we did not and do not argue that *E. coli* does not respond to histone expression in an adaptive manner, i.e. by activating or repressing genes that help it cope with the presence of histones. In fact, we previously highlighted downregulation of gyrases as a likely adaptive response. The key, from an evolutionary point of view, is that *E. coli* copes through activation of generic stress responses rather than through a mechanism specifically evolved to allow the accommodation of histones. The cells survive without machinery dedicated to dealing with histones or adaptations to the transcriptional or replicative apparatus.